# Human MAIT cell cytolytic effector proteins synergize to overcome carbapenem resistance in *Escherichia coli*

Caroline Boulouis[1◉], Wan Rong Sia[2◉], Muhammad Yaaseen Gulam[2], Jocelyn Qi Min Teo[3], Yi Tian Png[4], Thanh Kha Phan[5], Jeffrey Y. W. Mak[6,7], David P. Fairlie[6,7], Ivan K. H. Poon[6], Tse Hsien Koh[8], Peter Bergman[9], Chwee Ming Lim[4,10], Lin-Fa Wang[2], Andrea Lay Hoon Kwa[2,3‡], Johan K. Sandberg[1‡], Edwin Leeansyah[1,2,11]*

1 Center for Infectious Medicine, Department of Medicine, Karolinska Institutet, Stockholm, Sweden, 2 Programme in Emerging Infectious Diseases, Duke-National University of Singapore Medical School, Singapore, Singapore, 3 Department of Pharmacy, Singapore General Hospital, Singapore, Singapore, 4 Department of Otorhinolaryngology-Head and Neck Surgery, Singapore General Hospital, Singapore, Singapore, 5 Department of Biochemistry and Genetics, La Trobe Institute for Molecular Science, La Trobe University, Melbourne, Victoria, Australia, 6 Division of Chemistry and Structural Biology, Institute for Molecular Bioscience, The University of Queensland, Brisbane, Queensland, Australia, 7 Australian Research Council Centre of Excellence in Advanced Molecular Imaging, The University of Queensland, Brisbane, Queensland, Australia, 8 Department of Microbiology, Singapore General Hospital, Singapore, Singapore, 9 Department of Laboratory Medicine, Division of Clinical Microbiology, Karolinska Institutet, Stockholm, Sweden, 10 Surgery Academic Clinical Program, Duke-National University of Singapore Medical School, Singapore, Singapore, 11 Tsinghua-Berkeley Shenzhen Institute, Tsinghua University, Shenzhen, Peoples' Republic of China

◉ These authors contributed equally to this work.
‡ These authors also contributed equally to this work.
* edwin.leeansyah@ki.se

**Data Availability Statement:** All relevant data are within the paper and its Supporting Information files. All whole genome sequence of carbapenem-

## Abstract

Mucosa-associated invariant T (MAIT) cells are abundant antimicrobial T cells in humans and recognize antigens derived from the microbial riboflavin biosynthetic pathway presented by the MHC-Ib-related protein (MR1). However, the mechanisms responsible for MAIT cell antimicrobial activity are not fully understood, and the efficacy of these mechanisms against antibiotic resistant bacteria has not been explored. Here, we show that MAIT cells mediate MR1-restricted antimicrobial activity against *Escherichia coli* clinical strains in a manner dependent on the activity of cytolytic proteins but independent of production of pro-inflammatory cytokines or induction of apoptosis in infected cells. The combined action of the pore-forming antimicrobial protein granulysin and the serine protease granzyme B released in response to T cell receptor (TCR)-mediated recognition of MR1-presented antigen is essential to mediate control against both cell-associated and free-living, extracellular forms of *E. coli*. Furthermore, MAIT cell-mediated bacterial control extends to multidrug-resistant *E. coli* primary clinical isolates additionally resistant to carbapenems, a class of last resort antibiotics. Notably, high levels of granulysin and granzyme B in the MAIT cell secretomes directly damage bacterial cells by increasing their permeability, rendering initially resistant *E. coli* susceptible to the bactericidal activity of carbapenems. These findings define the role of cytolytic effector proteins in MAIT cell-mediated antimicrobial activity and indicate that

resistant E. coli files are available from the NCBI Bioproject database PRJNA577535 (https://www.ncbi.nlm.nih.gov/bioproject/) under accession numbers SRR10829621, SRR10829622, SRR10829623, and SRR10829624.

**Funding:** This research was supported by Swedish Research Council Grant 2015-00174, Marie Skłodowska Curie Actions, Cofund, Project INCA 600398, the Jonas Söderquist Foundation for Virology and Immunology, and the Petrus and Augusta Hedlund Foundation (EL). Further support came from the Swedish Research Council Grant 2016-03052, Swedish Cancer Society Grant CAN 2017/777, Center for Innovative Medicine Grant 20190732, and the National Institutes of Health Grant R01DK108350 (to JKS), as well as the CoSTAR-HS ARG Seed Fund 2018/02, NMRC Collaborative centre grant NMRC/CG/C005B/2017_SGH (to ALHK). CB is supported by the Karolinska Institutet Doctoral Grant and the Erik and Edith Fernström Foundation for Medical Research. PB is supported by a grant from the Swedish Research Council, the Stockholm County Council, Scandinavian Society for Antimicrobial Chemotherapy, The Swedish Foundation for Antimicrobial Resistance and the Karolinska Institutet. DPF acknowledges an ARC grant (CE140100011) and an NHMRC SPR Fellowship (1117017). The funders had no role in study design, data collection and analysis, decision to publish, or preparation of the manuscript.

**Abbreviations:** 293T-hMR1, 293T cells stably transfected with human MR1; 5-OP-RU, 5-(2-oxopropylideneamino)-6-D-ribitylaminouracil; Abs, antibodies; Ac-IETD-CHO, N-acetyl-L-isoleucyl-L-α-glutamyl-N-[(1S)-2-carboxy-1-formylethyl]-L-threoninamide trifluoroacetate; AMR, antimicrobial resistant; Casp, caspase; CRE, carbapenem-resistant Enterobacteriaceae; CREC, carbapenem-resistant *E. coli*; CTV, CellTrace Violet; Gnly, granulysin; Grz, Granzyme; GTL, GranToxiLux; IFNγ, interferon-γ; IL-17A, interleukin-17A; LB, Luria (lysogeny) broth; MACS, magnetic-activated cell sorting; MAIT, Mucosa-associated invariant T;

granulysin and granzyme B synergize to restore carbapenem bactericidal activity and overcome carbapenem resistance in *E. coli*.

## Introduction

Mucosa-associated invariant T (MAIT) cells are innate-like T cells that are highly abundant in mucosal tissues, the liver, lungs and gastrointestinal tract, and in peripheral blood (PB) [1]. MAIT cells are mostly CD8α$^+$ [2,3], express a semi-invariant T cell receptor (TCR), and recognize antigens in complex with the MHC-Ib-related protein (MR1) [4]. MR1 displays an extraordinary level of evolutionary conservation among placental and marsupial mammals [5], strongly supporting the notion that MR1 and MAIT cells perform critical functions in the immune system. The MR1-presented antigens recognized by MAIT cells are derivatives of intermediates in the microbial synthesis of vitamin B$_2$ (riboflavin) and are produced by many bacteria [6–8]. Riboflavin is a critical component in a wide variety of bacterial cellular processes [9]. MAIT cells are thus able to recognize and respond to a broad set of bacteria [10]. Following TCR-mediated recognition of MR1-presented bacterial riboflavin metabolite antigens, MAIT cells rapidly mediate a range of effector responses, including cytokine production, cytotoxicity, antimicrobial activity, and tissue repair function [11–18]. The abundance and antimicrobial features of MAIT cells, as well as the high evolutionary conservation of MR1, strongly suggest that MAIT cells are important for the protection of the host against bacterial pathogens [19]. Moreover, the conserved nature of the MAIT cell antigens across bacterial species via the shared components of the riboflavin biosynthetic pathway and antimicrobial features of MAIT cells support the hypothesis that they may have the capacity to recognize and respond to drug-resistant bacteria.

Infections caused by antimicrobial-resistant (AMR) bacteria are a serious threat to global public health. In particular, carbapenem-resistant Enterobacteriaceae (CRE), including *Escherichia coli*, *Klebsiella pneumoniae*, and *Enterobacter* spp., have emerged as a serious problem for hospitalized patients [20]. CRE are multidrug-resistant gram-negative bacteria that have acquired further resistance to one class of the last resort antibiotics, the carbapenems. Resistance to carbapenems in Enterobacteriaceae involves multiple mechanisms, including expression of efflux pumps, impermeability due to porin loss, and expression of β-lactamases with the ability to degrade carbapenems [20]. The polymyxins, and more specifically colistin, are the last resort antibiotics currently available to treat CRE infections. However, polymyxin-resistant CRE is on the rise, rendering them extensively drug-resistant (XDR) or even pan-drug resistant (PDR) [21]. The World Health Organization has therefore listed CRE in the critical category requiring further research and development of new treatments [22].

Despite the body of evidence that MAIT cells contribute to bacterial clearance and play a protective role in various bacterial infections [19], the mechanisms of MAIT cell antimicrobial activity remain relatively little explored. Moreover, it is unknown whether MAIT cell antimicrobial activity extends to AMR bacterial pathogens. In this study, we therefore investigated the mechanisms underlying MAIT cell antimicrobial activity by dissecting the roles of cytolytic proteins and cytokines in controlling bacterial growth. We extended these investigations by exploring the ability of MAIT cell cytolytic proteins, released in response to cognate recognition of MR1-presented antigen, to potentiate carbapenem-induced bactericidal activity against primary clinical isolates of *E. coli* from patients with CRE infections. Notably, MAIT cell antimicrobial activity mediated by the combined action of granulysin (Gnly) and Granzyme (Grz)

MIC, minimum inhibitory concentration; MR1, MHC-Ib-related protein; NAM, nafamostat mesylate; NP, nasopharyngeal; PB, peripheral blood; PBMC, peripheral blood mononuclear cell; PDR, pan-drug resistant; Prf, perforin; Pro-VAD-FMK, valyl-alanyl-aspartyl-[O-methyl]-fluoromethylketone; RAM, riboflavin assay medium; rh, recombinant human; RM, repeated measures; ROS, reactive oxygen species; RT, room temperature; TCR, T cell receptor; TNF, tumor necrosis factor; XDR, extensively drug-resistant.

B potently enhanced bactericidal activity of carbapenems against carbapenem-resistant strains of *E. coli*. Our findings thus define the mechanism underlying MAIT cell antimicrobial activity and support the concept that MAIT cell mobilization may protect the host from drug-resistant bacterial pathogens.

## Results

### MAIT cells kill *E. coli*–infected cells and reduce cell-associated bacterial loads in infected cells

To study the antimicrobial effector functions of MAIT cells, we developed an approach using the drug-sensitive *E. coli* clinical strain EC120S isolated from a patient suffering from a bloodstream infection [23]. We also developed a method for short-term PB MAIT cell expansion to provide the cell numbers with high purity required to perform such assays (S1A and S1B Fig). The *E. coli* strain EC120S infected HeLa cells efficiently as determined by pHrodo red-labeled live bacteria (S1C and S1D Fig). pHrodo becomes fluorescent in the acidic environment of cellular endosomal compartments and can therefore be used to evaluate *E. coli* internalization [24]. EC120S infected, replicated, and remained viable within HeLa cells' intracellular compartments for at least 24 h post-infection (S1E Fig). MAIT cells rapidly degranulated and killed the tested epithelial cell lines (HeLa and A549) infected with *E. coli* EC120S (Fig 1A–1D; S1F and S1G Fig) through an MR1-dependent mechanism (Fig 1C and 1D).

Because MAIT cells achieved complete lysis (Caspase [Casp]3$^+$ amine-reactive dead cell marker [DCM]$^+$) of the infected cells by 24 h, we examined MAIT cell antimicrobial activity after 3 h, when the cell membrane of most infected cells undergoing apoptosis remained intact, as shown by staining with amine-reactive cytoplasmic dyes (Fig 1B and 1C). Notably, bacterial viability inside both HeLa and A549 cells was significantly reduced in the presence of MAIT cells, via an MR1-dependent mechanism (Fig 1E, S1H Fig). To assess whether the reduced bacterial load reflected true bacterial killing by MAIT cells, or simply the release of bacteria into the supernatants during the assay, the bacterial loads in the cell lysates alone, or in lysates containing infected-cells and cell-free supernatants (total lysates), were enumerated. In both conditions, the presence of MAIT cells decreased bacterial load, indicating direct bacterial control (Fig 1F). Contrary to MAIT cells, Vα7.2$^-$ non-MAIT T cells were not able to control bacterial growth (S1I–S1K Fig). In addition, the capacity of MAIT cells to degranulate and induce apoptosis of infected target cells was superior to that of Vα7.2$^-$ T cells (S1I and S1J Fig). We next investigated whether MAIT cells at resting state were similarly able to mediate antimicrobial activity. Consistent with previous studies [24,25], resting MAIT cells purified using the Vα7.2 beads were inefficient in degranulation and killing of HeLa cells infected with *E. coli* (Fig 1G and 1H; S2A Fig) or pulsed with the positive control MR1 ligand 5-(2-oxopropylideneamino)-6-D-ribitylaminouracil (5-OP-RU; S2B and S2C Fig). Interestingly, resting MAIT cells also failed to control intracellular bacterial loads in *E. coli*–infected cells (Fig 1I). To determine whether the inability of resting MAIT cells to mediate antimicrobial activity was simply due to their weak cytotoxicity, purified MAIT cells were stimulated with IL-2 + IL-7 for 2 d to allow up-regulation of cytotoxic molecules [24,25]. These cytokine-activated MAIT cells were able to degranulate (Fig 1G; S2A Fig) and efficiently kill *E. coli*–infected cells (Fig 1H; S2A Fig). Similar or greater response was also observed following stimulation with the positive control MR1 ligand 5-OP-RU (S2B and S2C Fig). However, these newly activated MAIT cells still failed to efficiently control bacterial loads within infected cells, compared to the bacterial control exerted by MAIT cells that had been cultured for 15 d (Fig 1I). Interestingly, this temporal regulation of MAIT cell antimicrobial activity appeared to correspond to the differential

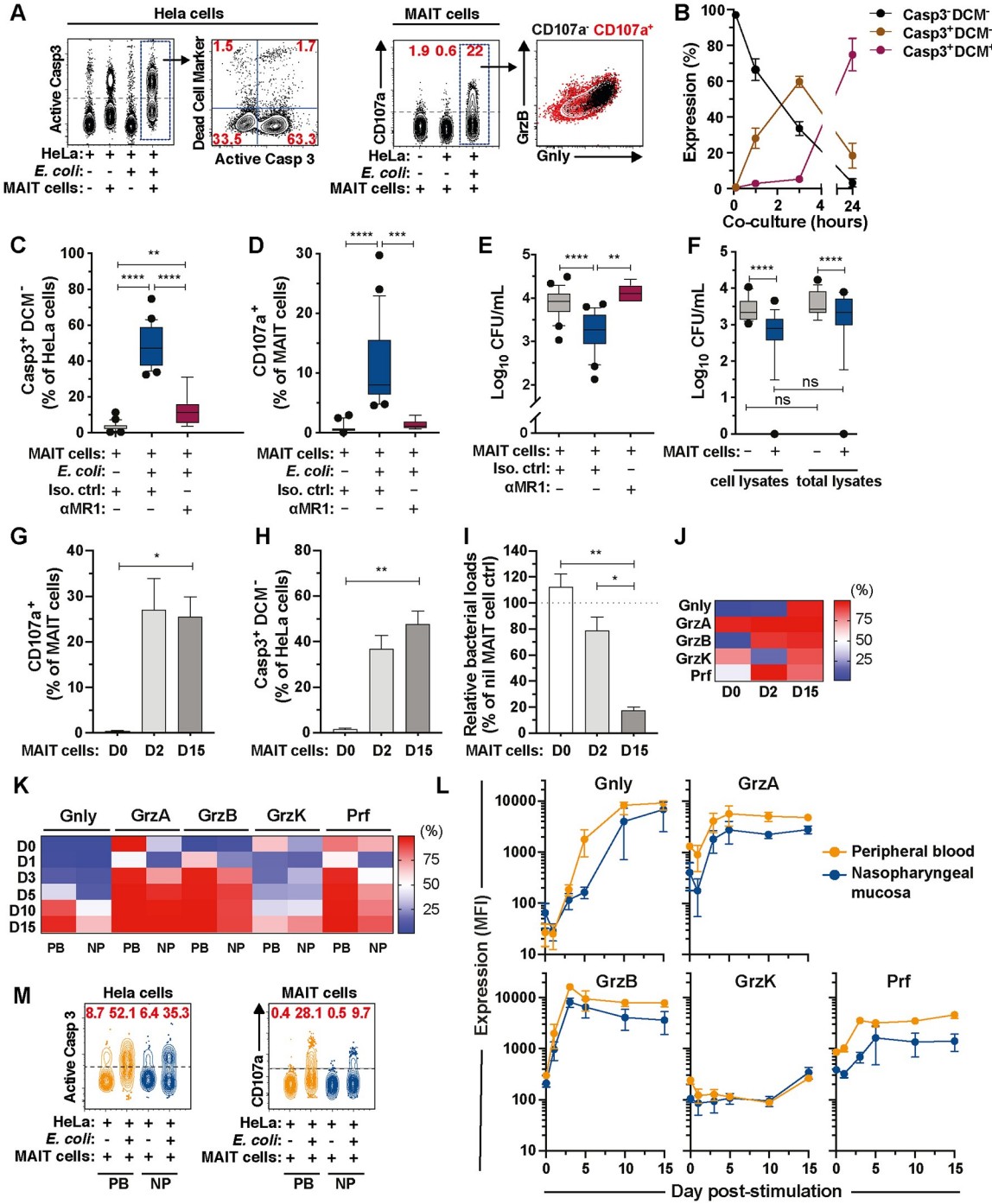

**Fig 1. MAIT cells kill bacteria-infected cells and suppress bacterial loads.** (A) Assessment of apoptosis of *E. coli* EC120S-infected HeLa cells by Casp3 activity and MAIT cell degranulation by CD107a, Gnly, and GrzB expression. (B) Apoptosis of infected HeLa cells at indicated time point ($n$ = 9–11). (C) Measurement of early apoptosis (Casp3$^+$DCM$^-$) on uninfected or EC120S-infected HeLa cells, (D) degranulation by MAIT cells, and (E) bacterial counts following lysis of infected HeLa cells after 3 h of co-culture with and without MAIT cells in the presence of anti-MR1 or isotype control ($n$ = 7–8 for EC120S-infected HeLa cells+anti-MR1, $n$ = 24–25 for others). (F) Bacterial counts in infected HeLa cell lysates or in total lysates (cell lysates plus supernatants) after 3 h of co-culture with or without MAIT cells ($n$ = 14). (G) MAIT cell degranulation, (H) apoptosis of EC120S-infected HeLa cells, and (I) relative bacterial loads following co-culture of EC120S-infected HeLa cells with MAIT cells derived from D0, 2, and 15 of culture ($n$ = 4). (J) Percentage of cytolytic proteins expressed by MAIT cells from D0, 2, and 15 of culture assessed by flow cytometry ($n$ = 3–10). (K, L) Flow cytometry analysis on frequency (K) and levels (MFI) (L) of cytolytic protein expression by PB ($n$ = 4) and NP ($n$ = 3) MAIT cells at various time points. (M) Representative FACS plot of apoptosis of EC120S-infected HeLa cells and MAIT cell degranulation following co-culture with PB or NP

MAIT cells ($n = 3$). Data presented as heat map shows the mean, whereas data presented as line or bar graphs with error bars represent the mean and standard error. Box and whisker plots show median, the 10th to 90th percentile, and the interquartile range. Statistical significance was calculated using mixed-effects analysis followed by Tukey's multiple comparison test (C–E), Wilcoxon's signed-rank test or Friedman's test with Dunn's multiple comparisons test (F), or repeated-measure one-way ANOVA (G–I). $^{****}p < 0.0001$, $^{***}p < 0.001$, $^{**}p < 0.01$, $^{*}p < 0.05$. The underlying data of this figure can be found in S1 Data. Casp, caspase; CFU, colony-forming units; D, day; DCM, dead cell marker; FACS, fluorescence-activated cell sorting; Gnly, granulysin; GrzB, Granzyme B; MAIT, Mucosa-associated invariant T; MFI, mean fluorescence intensisty; MR1, MHC-Ib-related protein; NP, nasopharyngeal; ns, not significant; PB, peripheral blood.

expression of cytolytic proteins by MAIT cells (Fig 1J; S2D and S2E Fig). These results indicate that antigen-activated MAIT cells have the capacity to kill infected target cells in an MR1-dependent manner through Casp3 activation and progressively over several days develop the ability to control bacterial growth within infected cells.

## Tissue-resident MAIT cells acquire cytolytic proteins with similar kinetics but of lower magnitudes

MAIT cells are abundant in the mucosal surfaces that serve as barriers to the external microbe-rich environments. To investigate the cytolytic capacity of MAIT cells in mucosa, tissue-resident MAIT cells were obtained from the nasopharyngeal (NP) mucosa of healthy individuals undergoing nasal polyp removals. CD69$^+$ CD103$^+$ tissue-resident NP MAIT cells (S2F Fig) overall expressed a similar set of cytolytic proteins, although at somewhat lower proportion and magnitude of cytolytic proteins at baseline and following in vitro culture when compared to those of matched PB MAIT cells (Fig 1K and 1L). However, the kinetics of cytolytic protein acquisition in NP MAIT cells appeared to be comparable to that of PB MAIT cells (Fig 1L; S2G Fig). Finally, NP MAIT cells degranulated and killed HeLa cells infected with *E. coli* EC120S (Fig 1M). These findings suggest that tissue-resident MAIT cells are able to mount cytotoxicity against bacteria-infected cells within the mucosal tissue environment.

## MAIT cells mediate antimicrobial activity through the cytolytic protein-dependent pathway

Next, using PB MAIT cells, the role of cytolytic proteins in MAIT cell antimicrobial activity against cell-associated form of *E. coli* was investigated. Initially, the intracellular levels of GrzA, GrzB, GrzK, Gnly, and perforin (Prf) were measured using flow cytometry to determine the transfer of cytolytic effector proteins into *E. coli*–infected cells after co-culture with MAIT cells. With the exception of GrzK, all measured cytolytic proteins were detected in HeLa target cells (Fig 2A), matching their expression pattern in MAIT cells (Fig 2A; S2D and S2E Fig). The inhibition of cell-associated bacterial growth associated negatively with cell viability (Fig 2B) but positively with apoptosis (Fig 2C) and MAIT cell expression of GrzB (Fig 2D and 2E), Gnly (Fig 2F), and co-expression of GrzB and Gnly (Fig 2G). Because GrzB was readily detected inside *E. coli*–infected target cells co-cultured with MAIT cells (Fig 2A), we evaluated GrzB activity within the infected target cells using a fluorescent GrzB substrate (GranToxiLux) [26]. Active GrzB was detected in HeLa target cells infected with *E. coli* (Fig 2H), indicating that MAIT cells delivered GrzB into infected cells.

In order to assess MAIT cell use of the cytolytic protein pathway to control cell-associated bacterial growth, cytolytic protein activity was selectively inhibited by pharmacological inhibitors. Ethylene glycol tetraacetic acid (EGTA), a Ca$^{2+}$-specific chelator that inhibits the release of cytolytic granules, was used in the presence of Mg$^{2+}$ supplementation. EGTA strongly decreased MAIT cell degranulation as determined by CD107a expression (Fig 2I and 2J),

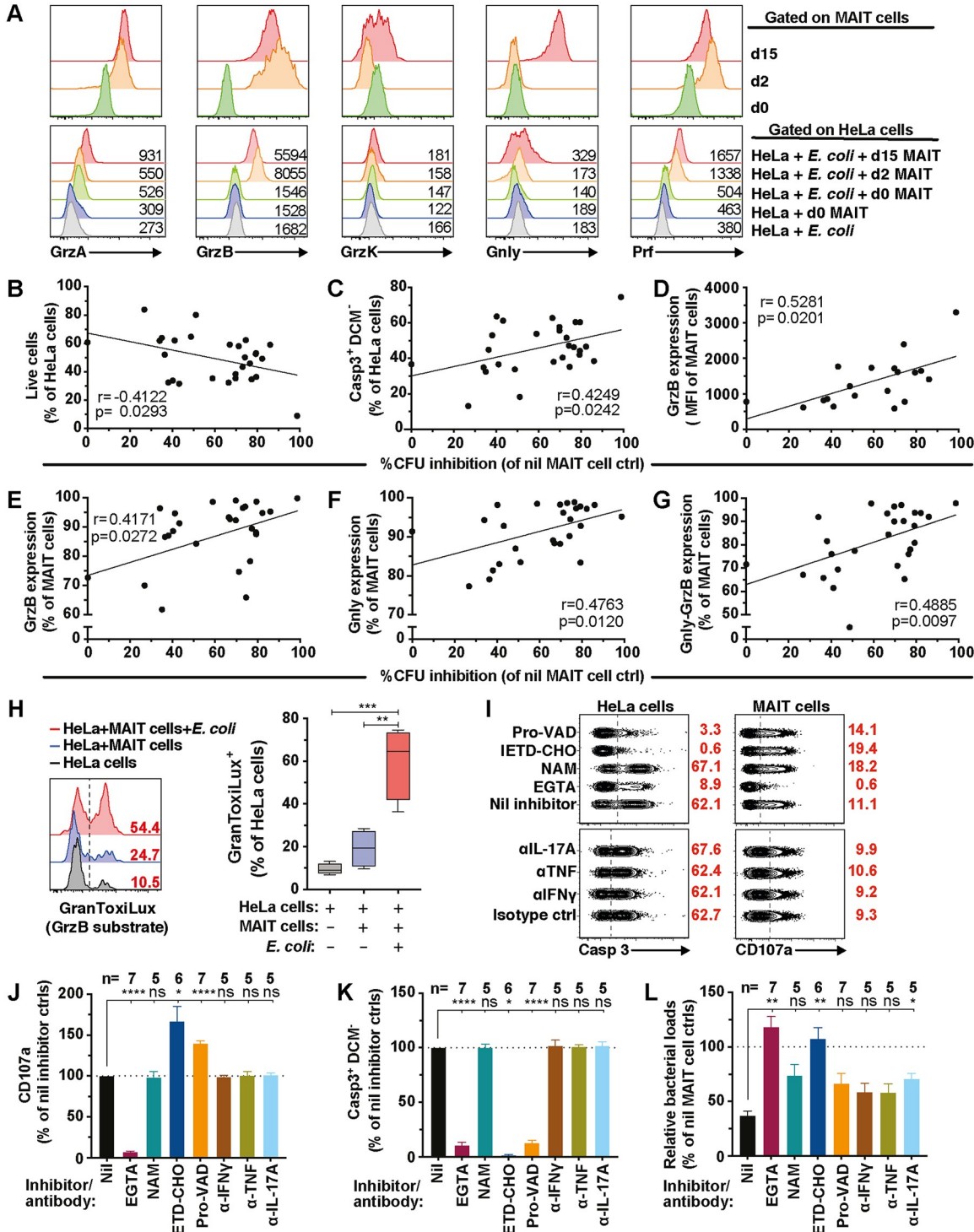

**Fig 2. MAIT cell antimicrobial activity is associated with cytolytic protein expression.** (A) Detection of cytolytic protein contents in effector MAIT cells and target *E. coli* EC120S-infected or uninfected HeLa cells following 3 h co-culture with MAIT cells obtained at different days following MAIT cell activation. Representative histograms from 4 independent donors are shown. (B) Correlation between live ($n = 28$) and (C) apoptotic ($n = 28$) EC120S-infected HeLa cells, (D) GrzB intensity ($n = 19$), (E) GrzB expression ($n = 28$), (F) Gnly expression ($n = 27$), and (G) Gnly-GrzB co-expression ($n = 27$) in MAIT cells with the inhibition of bacterial growth. (H) Levels of the GrzB substrate GranToxiLux activity in uninfected or EC120S-infected HeLa cells with or without MAIT cells ($n = 4$). (I) Flow cytometry plots of (J) degranulation by MAIT cells, (K) apoptosis in EC120S-infected HeLa cells, and (L) the relative bacterial loads in the presence of the indicated inhibitors or mAbs. Bar graphs with error bars represent the mean and standard error. Box and whisker

plots show median, the 10th to 90th percentile, and the interquartile range. Statistical significance was determined using the ordinary one-way ANOVA (H), or mixed-effects analysis (J–L) followed by Dunnett's or Tukey's post hoc test as appropriate. Correlations were calculated using the Pearson (B, C) or the Spearman test (D–G). $****p < 0.0001$, $***p < 0.001$, $**p < 0.01$, $*p < 0.05$. The underlying data of this figure can be found in S1 Data. α, anti; Casp, caspase; d, day; ctrl, control; EGTA, ethylene glycol tetraacetic acid; Gnly, granulysin; Grz, Granzyme; IETD-CHO, N-acetyl-L-isoleucyl-L-α-glutamyl-N-[(1S)-2-carboxy-1-formylethyl]-L-threoninamide trifluoroacetate; mAbs, monoclonal antibodies; MAIT, Mucosa-associated invariant T; Nil, untreared; ns, not significant; Prf, perforin; Pro-VAD, valyl-alanyl-aspartyl-[O-methyl]-fluoromethylketone.

reduced the delivery of cytolytic proteins (S2H Fig) and Casp3 activation in infected target cells (Fig 2I and 2K), and abolished the bacterial control by MAIT cells (Fig 2L). To investigate the role of GrzA and GrzB in MAIT cell antimicrobial activity, their activation was blocked using nafamostat mesylate (NAM) and N-acetyl-L-isoleucyl-L-α-glutamyl-N-[(1S)-2-carboxy-1-formylethyl]-L-threoninamide trifluoroacetate (Ac-IETD-CHO), respectively [27,28]. NAM had no effect on MAIT cell degranulation, Casp3 activation, and bacterial counts (Fig 2I–2L). In contrast, although Ac-IETD-CHO had no effect on MAIT cell degranulation (Fig 2I and 2J), it strongly abrogated Casp3 activation in infected target cells (Fig 2I and 2K) and impaired bacterial control (Fig 2L). To assess whether apoptosis of infected cells also contributed to bacterial control, Casp3 activation was blocked using the pan-Casp inhibitor valyl-alanyl-aspartyl-[O-methyl]-fluoromethylketone (Pro-VAD-FMK). The pan-Casp inhibitor had no effect on MAIT cell degranulation (Fig 2I and 2J) but severely diminished Casp3 activation in infected cells without affecting bacterial counts (Fig 2I–2L). This suggests that bacterial control by MAIT cells was independent of infected cell apoptosis. Finally, we assessed whether interferon (IFN)γ, tumour necrosis factor (TNF), and interleukin (IL)-17A, pro-inflammatory cytokines produced by MAIT cells (S2I Fig), played any role in MAIT cell bacterial control by blocking with neutralizing antibodies (Abs). Blocking these cytokines did not significantly affect degranulation or Casp3 activation (Fig 2I–2K). IL-17A blockade, but not IFNγ or TNF blockade, only slightly diminished MAIT cell antimicrobial activity (Fig 2L). Altogether, these findings indicate that MAIT cells use the cytolytic protein pathway to kill bacterially infected cells, and this may play an important role in early control of cell-associated bacterial loads.

## MAIT cells recognize and mediate antimicrobial activity against carbapenem-resistant *E. coli* clinical strains

Next, we tested the hypothesis that MAIT cells maintain antimicrobial activity against carbapenem-resistant *E. coli* (CREC) clinical isolates. To address this, we first investigated whether CREC strains EC234, EC241, EC362, and EC385 (S1 Table) were riboflavin autotrophs by culture in riboflavin assay medium (RAM), a riboflavin-deficient broth. All CREC isolates grew in both RAM and nutrient-rich lysogeny broth (LB) at comparable rates, and the addition of external riboflavin did not influence their growth (S3A–S3H Fig). Moreover, the expression of *ribA*, the gene encoding the first enzyme of the riboflavin biosynthesis pathway, was similar among the clinical isolates (S3I Fig). These findings confirmed that the CREC clinical isolates were riboflavin-synthesis competent, a requirement for generating the riboflavin-related bacterial metabolite antigens [7].

Next, the ability of CREC to stimulate MAIT cells was tested by incubating peripheral blood mononuclear cells (PBMCs) with fixed bacteria for 24 h. All CREC strains induced degranulation and expression of GrzB, IFNγ, TNF, and IL-17A by MAIT cells at comparable levels (Fig 3A; S3J Fig). Furthermore, MAIT cells displayed similar patterns of polyfunctionality against these diverse strains, although some differences in response profiles were noted (S3K Fig). These variations occurred despite similar uptake of the bacteria by the PBMCs

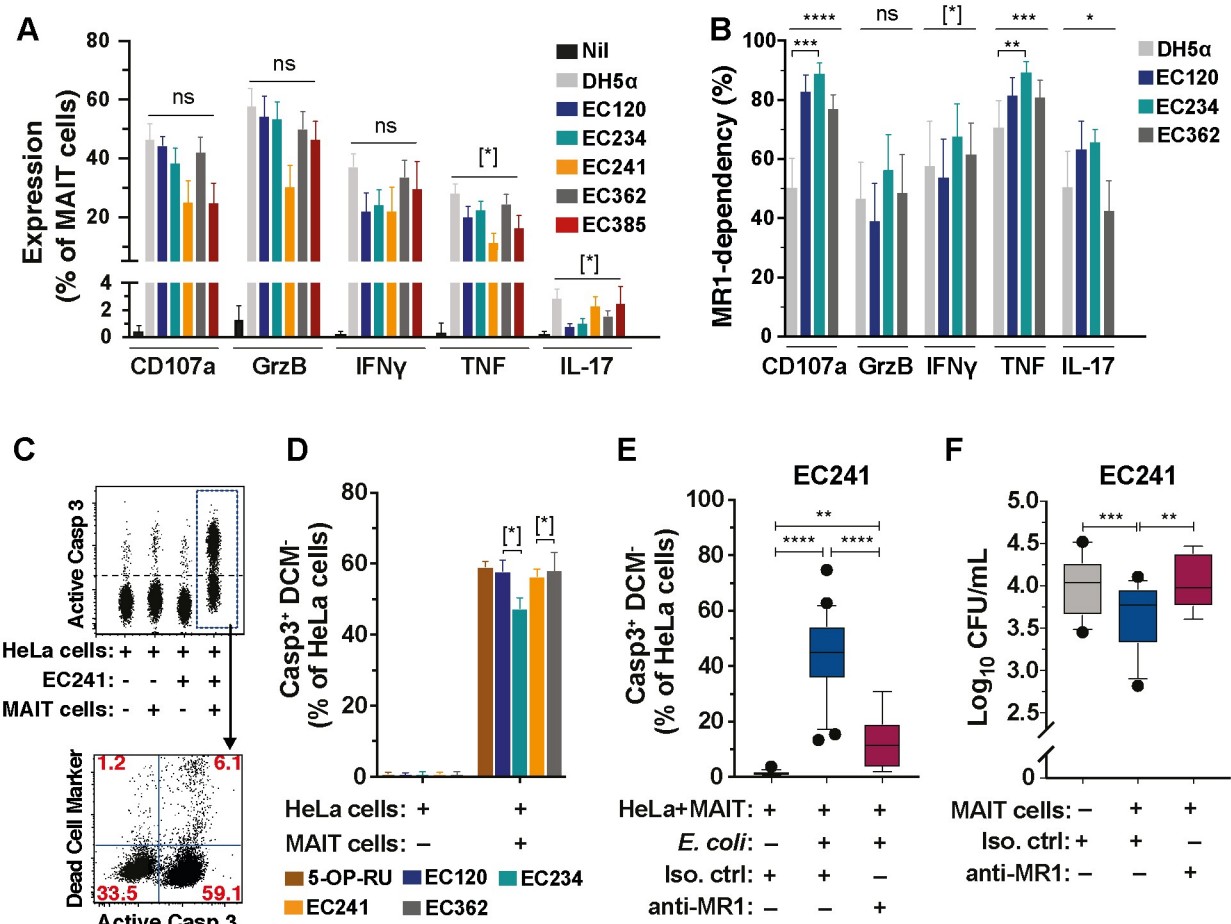

**Fig 3. MAIT cells respond to and control CREC.** (A) Expression of CD107a, GrzB, IFNγ, TNF, and IL-17A in MAIT cells stimulated for 24 h with *E. coli* strains DH5α (*n* = 16), EC120S (*n* = 7), and the carbapenem-resistant strains EC234 (*n* = 16), EC241 (*n* = 5), EC362 (*n* = 16), and EC385 (*n* = 5). Unstimulated, *n* = 16. (B) MR1-dependency of effector protein and cytokine production by MAIT cells stimulated with indicated strains. MR1-dependency was calculated as previously described [2] (*n* = 7). (C) Representative flow cytometry plots of Casp3 activation and apoptosis in HeLa cells alone, HeLa cells infected with EC241, or co-cultured with MAIT cells with or without EC241 for 3 h. (D) Apoptosis of HeLa cells alone or co-cultured with MAIT cells for 3 h in the presence of 5-OP-RU, EC120S, EC234, EC241, or EC362 (*n* = 5). (E, F) Casp3 activation (E) and bacterial loads (F) in HeLa cells infected with EC241 co-cultured with MAIT cells in the presence of anti-MR1 antibody or isotype control (*n* = 5–8 for EC241-infected HeLa cells+anti-MR1 mAb, *n* = 13–25 others). Data presented as bar graphs with error bars represent the mean and standard error. Box and whisker plots show median, the 10th to 90th percentile, and the interquartile range. Statistical significance was determined using the Kruskal-Wallis ANOVA (A) or the Friedman test (B, D) followed by Dunn's multiple comparison test, or mixed-effects analysis followed by Dunnett's multiple comparison test (E, F). ****$p < 0.0001$, ***$p < 0.001$,**$p < 0.01$,*$p < 0.05$, [*]$p < 0.1$. The underlying data of this figure can be found in S1 Data. Casp, caspase; CFU, colony-forming units; CREC, carbapenem-resistant *E. coli*; Ctrl, control; DCM, dead cell marker; FCS-A, forward scatter area; Grz, Granzyme; IFNγ, interferon γ; IL-17A, interleukin-17A; mAb, monoclonal antibody; MAIT, Mucosa-associated invariant T; MR1, MHC-Ib-related protein; TNF, tumor necrosis factor; 5-OP-RU, 5-(2-oxopropylideneamino)-6-D-ribitylaminouracil.

(S3L Fig). MAIT cell responses to CREC were predominantly MR1-dependent although this differed somewhat between the strains (Fig 3B).

To assess the capacity of MAIT cells to kill CREC-pulsed cells, HeLa or 293T-hMR1 cells were pulsed with strains EC234, EC241, EC362, or with the drug-sensitive EC120S. In all cases, MAIT cells recognized and killed HeLa cells (Fig 3C and 3D) and 293T cells stably transfected with human MR1 (293T-hMR1) (S3M–S3O Fig) pulsed with CREC strains at a comparable levels, though some variations were also noted. To investigate whether MAIT cells can mediate antimicrobial activity against CREC strains, we selected strain EC241 because it most efficiently enters and is internalized by HeLa cells (S3P Fig). MAIT cells induced Casp3

activation in HeLa cells infected with EC241 in an MR1-dependent manner (Fig 3E), consistent with results using the drug-sensitive strain EC120S (Fig 1). Furthermore, MAIT cells significantly reduced the CREC cell-associated bacterial loads in an MR1-dependent fashion (Fig 3F). Taken together, MAIT cells recognize CREC, mediate killing of infected cells, and reduce cell-associated bacterial loads in an MR1-dependent manner.

## MAIT cells secrete high levels of cytolytic proteins and mediate antimicrobial activity against free-living CREC in the surrounding milieu

We next investigated whether MAIT cells release cytolytic proteins into the surrounding milieu in response to MR1-restricted cognate recognition of cells infected with a riboflavin-synthesis competent strain of *E. coli*. MAIT cells secreted high levels of GrzA, GrzB, and Gnly into the supernatants already after a short 3 h stimulation with *E. coli*–infected cells (Fig 4A). Because some cytolytic proteins have direct antimicrobial activity [29,30], the MAIT cell secretome ability to inhibit free-living, extracellular *E. coli* growth was investigated. To collect bacteria-free MAIT cell secretomes following TCR-stimulation, 293T-hMR1 cells were used as antigen-presenting cells and pulsed with the synthetic antigen 5-OP-RU [31] and co-cultured with Vα7.2 bead-purified MAIT cells (S4A Fig). Live *E. coli* strains were then incubated with supernatants obtained from these co-cultures (S4A Fig). MAIT cells strongly degranulated their cytolytic protein content following MR1-restricted TCR triggering, and these proteins accumulated at high levels in the supernatants after 24 h of co-culture (S4B and S4C Fig). To assess whether MAIT cell secretomes induce damage on extracellular bacteria, the cell-impermeable nucleic acid dye SYTOX Green was used [32] (S4D Fig). To allow detection of bacteria versus debris by flow cytometry, bacteria were prestained with the cell permeable SYTO 62 just before fixation (Fig 4B). In the presence of MAIT cell supernatants, the CREC strains EC234 and EC362 suffered an increase in membrane permeability, as seen by the significant increase of SYTOX Green and SYTO 62 staining when compared to the control supernatants (Fig 4B and 4C, S4E and S4F Fig). Finally, the bacterial counts of the *E. coli* strains exposed to the MAIT cell secretomes were significantly reduced, although not totally suppressed (Fig 4D). These results indicate that the MAIT cell secretomes display antimicrobial activity against extracellular *E. coli* by markedly increasing bacterial cell permeability.

To determine whether the suppression of extracellular bacteria was due to a bacteriostatic effect or true killing of the bacteria, a time-kill analysis [33] of live bacteria was performed using CREC strain EC362 with various starting inocula (S4G Fig). At the original high starting inoculum of $10^5$ CFU/mL, MAIT cell secretomes mildly but significantly inhibited the growth of EC362 at 2 h (S4G Fig), in agreement with the previous findings. However, the bacteria rapidly rebounded and grew over a 24 h period (S4G Fig). Interestingly, when the starting inocula were reduced to $10^4$ and $10^3$ CFU/mL, MAIT cell secretomes displayed stronger bacterial growth inhibition and true bacterial killing, as reflected by the drop in bacterial counts, lasting longer with lower starting inocula (S4G Fig). In summary, these findings suggest that the antimicrobial activity of MAIT cell secretomes against extracellular bacteria in this static culture system was influenced by the starting inoculum effect.

## MAIT cell secretomes enhance the bactericidal activity of carbapenems against extracellular CREC

Following the observation that MAIT cell secretomes can directly alter bacterial membrane permeability (Fig 4B–4D), we asked whether the MAIT cell secretome could enhance the bactericidal activity of carbapenems against CREC strains. To test this, CREC strains EC234 and EC362 were incubated in the presence of MAIT cell supernatants with titrated concentrations

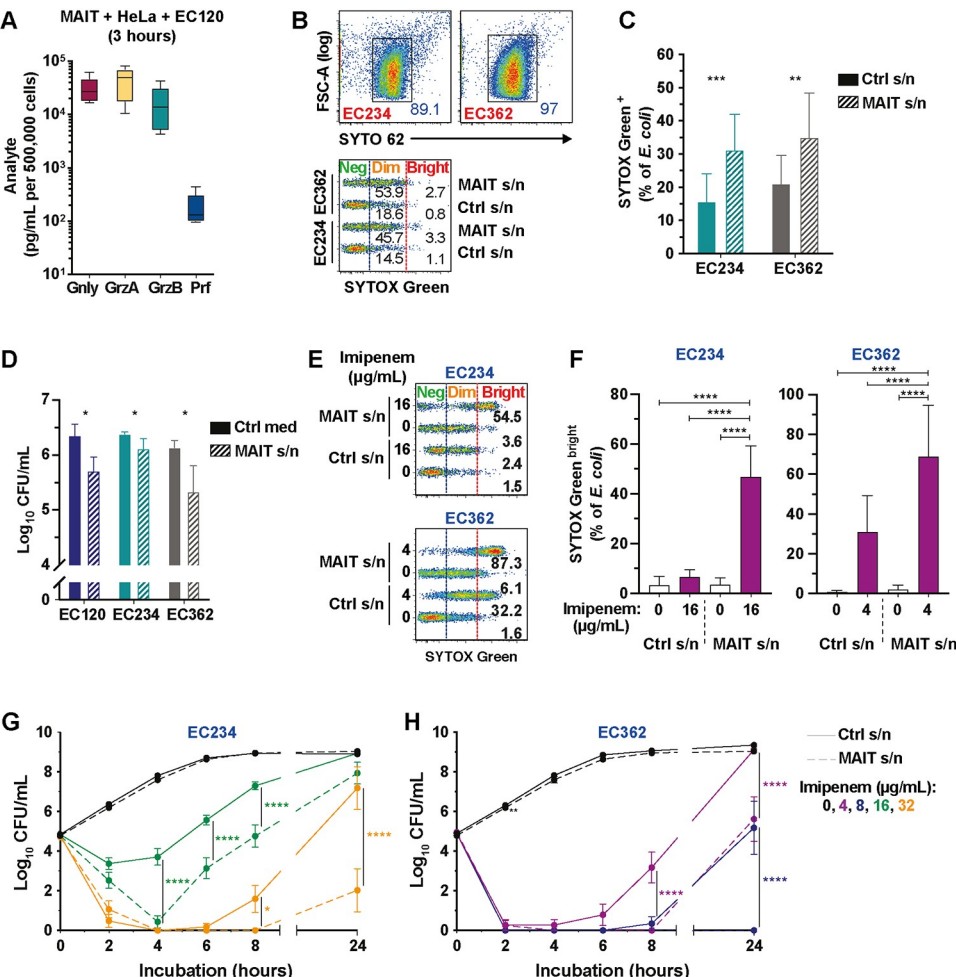

**Fig 4. MAIT cell secretomes mediate antimicrobial activity against CREC.** (A) Concentration of cytolytic proteins in the supernatant of MAIT cells following 3 h co-culture with *E. coli* EC120S-infected HeLa cells (*n* = 5–6). (B) Gating strategy for *E. coli* identification and quantification of SYTOX Green by flow cytometry. (C) SYTOX Green levels (*n* = 10) and (D) bacterial counts of various *E. coli* strains after incubation with MAIT cell or control supernatants for 2 h (*n* = 6 [EC120S], 12 [EC234, EC362]). (E, F) Levels of SYTOX Green^bright in *E. coli* EC234 and EC362 in the presence of MAIT cell or control supernatants supplemented with imipenem for 2 h (*n* = 10) or (G, H) the live bacterial counts over 24 h (*n* = 11–16 [EC234], 11–13 [EC362]). Data presented as line or bar graphs with error bars represent the mean and standard error. Statistical significance was determined using paired *t* test (C), Wilcoxon's signed-rank test (D), RM one-way ANOVA with Dunnett's post hoc test (F), and mixed-effects analysis with Sidak's post hoc test (G, H). Significant differences in bacterial counts cultured in control versus MAIT cell supernatants at indicated time points and imipenem concentrations in (G) and (H) were indicated by asterisks. ****$p < 0.0001$, ***$p < 0.001$, **$p < 0.01$, *$p < 0.05$. The underlying data of this figure can be found in S1 Data. CFU, colony-forming units; CREC, carbapenem-resistant *E. coli*; Ctrl, control; Gnly, granulysin; Grz, Granzyme; MAIT, Mucosa-associated invariant T; Prf, perfori; s/n, supernatant.

of the carbapenem antibiotics imipenem, ertapenem, and meropenem. Greatly enhanced permeability and damage to the bacteria was evident by the appearance of bacteria stained with high intensities for SYTOX Green (SYTOX Green^bright), in comparison with bacteria cultured in either MAIT cell supernatants or carbapenem antibiotics alone (Fig 4E and 4F; S4H and S4I Fig). Fluorescence intensity of SYTO 62 was also increased in bacteria treated with a combination of MAIT cell secretome and imipenem (S4J and S4K Fig).

Next, effects on the live bacterial counts were evaluated using a time-kill method [33]. Notably, MAIT cell secretomes in combination with imipenem killed the highly carbapenem-resistant *E. coli* strain EC234 (minimum inhibitory concentrations [MICs] to all carbapenems ≥32 μg/mL; S2 Table), as well as the XDR strain EC362 harboring further resistance to colistin (S1 Table). In both cases, imipenem concentrations were well below the strains' respective MICs (Fig 4G and 4H). Growth rates of strains EC234 and EC362 cultured in MAIT cell supernatants in the presence of titrated concentrations of imipenem, were significantly slower compared to their respective controls (S5A and S5B Fig). These growth delays occurred at imipenem concentrations up to 4-fold lower than the MIC for EC234 and up to 16-fold lower than the MIC for EC362 (S5A and S5B Fig). Moreover, the MAIT cell secretome decreased the imipenem concentrations required to fully suppress the growth of strains EC234 and EC362 by at least 2- and 4-fold, respectively (S5C Fig). In the presence of imipenem, the antimicrobial activity of MAIT cell secretomes appeared to be superior to that of non-MAIT Vα7.2⁻ T cells from the same donors (S5D Fig). Finally, we evaluated whether the antimicrobial activity was derived from the MAIT cells themselves or from the dying 293T-hMR1 cells used as antigen-presenting cells in the co-culture system. Firstly, supernatants of Vα7.2 bead-purified and expanded MAIT cells cultured without 293T-hMR1 cells still displayed antimicrobial activity in the presence of imipenem against the XDR strain EC362 (S5E and S5F Fig). Secondly, blocking the apoptosis of 293T-hMR1 cells caused by MAIT cell cytotoxicity with the pan-Casp inhibitor Pro-VAD-FMK caused no significant loss of antimicrobial activity in the presence of imipenem against strains EC234 and EC362 (S5G and S5H Fig). Taken together, these findings suggest that there is a synergy between MAIT cell secretome antimicrobial activity and carbapenems, thereby enhancing the in vitro bactericidal activity of carbapenems against extracellular CREC strains.

## MAIT cell secretome antimicrobial activity correlates with the levels of secreted cytolytic proteins

We next revisited the time-kill data (Fig 4G and 4H) and analyzed possible associations between the amounts of cytolytic proteins present in the MAIT cell supernatants and the antimicrobial activity when combined with imipenem. In the bacterial cultures in which growth was detected, there was significantly less Gnly and GrzB in the MAIT cell secretome (Fig 5A and 5B). However, no significant difference in GrzA and Prf levels was observed (Fig 5A and 5B). Moreover, there were negative correlations between viable bacterial counts and Gnly or GrzB levels within the MAIT cell secretomes (Fig 5C–5F) but no correlations with the levels of GrzA or Prf (S6A–S6D Fig). Interestingly, Gnly levels in the supernatants correlated positively with the length of the lag-phase for strains EC234 and EC362 in the presence of imipenem (S6E and S6F Fig). Furthermore, the MAIT cell secretomes that contained the highest concentrations of Gnly had significantly greater and longer inhibition of EC234 and EC362 (S6G Fig). Taken together, these findings suggest that cytolytic proteins, particularly Gnly and GrzB, may significantly contribute to the MAIT cell secretome antimicrobial activity against extracellular CREC in the presence of imipenem.

## Gnly and GrzB contribute to the antimicrobial activity of MAIT cell secretomes and potentiate the bactericidal action of carbapenems against CREC

Finally, the functional contribution of different cytolytic proteins to the MAIT cell secretome antimicrobial activity against CREC was examined. To enable this, the assay dynamic range was enhanced by using strain EC362, which was more sensitive to the inhibitory effects of the

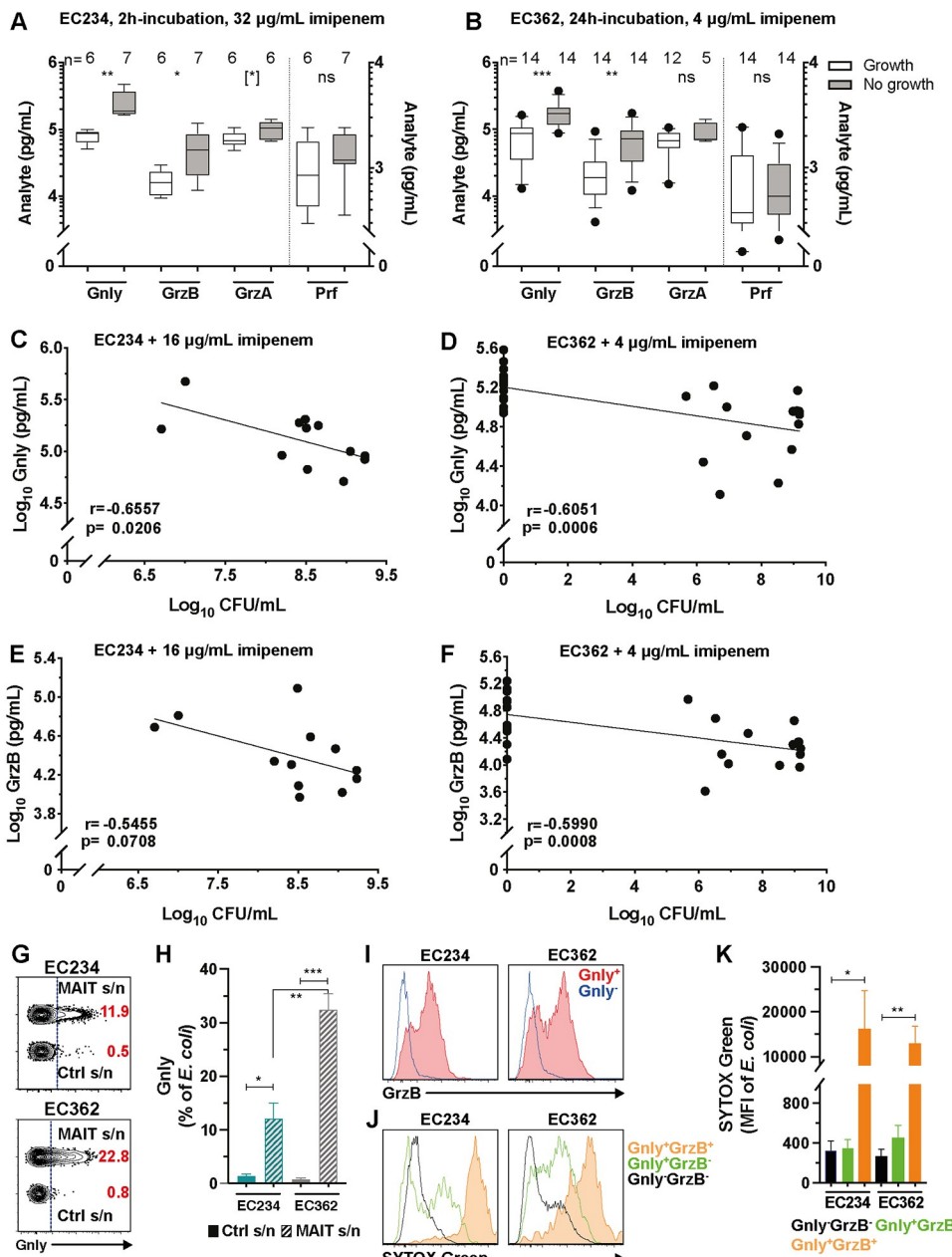

**Fig 5. Cytolytic proteins contribute to the antimicrobial activity of MAIT cell secretomes.** (A, B) Concentration of cytolytic proteins in the MAIT cell supernatants spiked with imipenem in the *E. coli* EC234 ($n = 12$) and EC362 ($n = 28$) cultures in which growth was detected or not at the indicated time points. (C, D) Correlation between Gnly and (E, F) GrzB levels in the MAIT cell supernatants and *E. coli* EC234 and EC362 bacterial loads after 24 h incubation in the MAIT cell supernatants spiked with imipenem. (G) Gating strategy of Gnly flow cytometry staining and (H) proportion of Gnly in *E. coli* EC234 ($n = 4$) and EC362 ($n = 6$) following 30 min incubation with MAIT cell or control supernatants. (I) Representative histograms of GrzB, and (J) SYTOX Green staining and (K) intensity in *E. coli* EC234 ($n = 4$) and EC362 ($n = 6$) following 30 min incubation in MAIT cell or control supernatants. Data presented as bar graphs with error bars represent the mean and standard error. Box and whisker plots show median, the 10th to 90th percentile, and the interquartile range. Statistical significance was determined using the Mann-Whitney test (A, B), paired *t* test for intrastrain or unpaired *t* test for interstrain analyses (H), and Friedman's test with Dunn's post hoc test (K). Correlations were calculated using the Pearson test (C, E) and the Spearman test (D, F). ***$p < 0.001$, **$p < 0.01$, *$p < 0.05$, [*] $p < 0.1$. The underlying data of this figure can be found in S1 Data. CFU, colony-forming units; Gnly, granulysin; Grz, Granzyme; MAIT, Mucosa-associated invariant T; MFI, mean fluorescence intensity; ns, not significant; Prf, perforin; s/n, supernatant.

MAIT cell secretomes (Fig 4H), and lower concentration of imipenem (2 μg/mL) to reduce its residual inhibitory effects. Blocking MAIT cell degranulation using EGTA + Mg$^{2+}$ (S6H Fig) nearly completely abolished the MAIT cell secretome-mediated bacterial inhibition (S6I Fig), whereas blocking GrzB activity alone had no observable effect (S6I Fig). Similarly, blocking the activity of IFNγ, TNF, or IL-17A during the MAIT-293T-hMR1 cell co-culture period had no significant effect on the antimicrobial activity of harvested supernatants (S6I Fig). Importantly, specific depletion of Gnly from the supernatants (S6J Fig) significantly reduced the MAIT cell secretome inhibition of bacterial growth (S6K Fig), as well as the capacity to induce bacterial cell permeability (S6L Fig). Gnly was detected intrabacterially by flow cytometry in both EC234 and EC362 incubated with MAIT cell supernatants (Fig 5G and 5H, S7A Fig). GrzB was also detected, but mostly in Gnly-containing bacteria (Fig 5I), whereas no GrzA or Prf was detected inside bacterial cells (S7A and S7B Fig). Notably, bacterial cells containing both Gnly and GrzB had higher membrane permeability than those containing only Gnly (Fig 5J and 5K). Short-term 30 min concomitant treatment with imipenem did not increase Gnly levels inside bacterial cells despite the increase in bacterial permeability and damage (S7C Fig). This suggests that the increase in bacterial death following incubation with MAIT cell supernatants and imipenem (Fig 4H) was possibly caused by the higher imipenem penetration into bacterial cells following MAIT cell secretome-mediated increase in bacterial cell permeability. Collectively, these findings indicate that Gnly and GrzB contribute to the antimicrobial activity of MAIT cell secretomes against extracellular *E. coli* and potentiate the bactericidal action of carbapenem antibiotics against CREC (Fig 6).

## Discussion

Despite a considerable body of evidence that MAIT cells play a protective role in various bacterial infections, the ability of MAIT cells to directly inhibit bacterial growth and the mechanisms mediating antimicrobial activity are little known. Furthermore, whether MAIT cells have antimicrobial activity against drug-resistant bacteria has not been studied. Here, we show that human MAIT cells have direct antimicrobial activity against cell-associated *E. coli* primary clinical isolates, including CREC strains that are extensively resistant to carbapenems. This antimicrobial activity depends on the TCR-mediated activation of cytolytic protein secretion by MAIT cells in response to cognate recognition of cells that were infected or have taken up bacteria and presented antigen. Interestingly, the same antimicrobial mechanism of MAIT cells is also acting against extracellular *E. coli*. High levels of cytolytic effector proteins secreted by MAIT cells into the surrounding milieu directly damage free-living *E. coli* bacterial cells by increasing their permeability, including those of CREC clinical isolates. Strikingly, MAIT cell secretomes restore the bactericidal activity of carbapenems against free-living CREC clinical isolates in vitro. The levels of cytolytic proteins secreted by MAIT cells, in particular Gnly and GrzB, correlate with the degree of extracellular bacterial killing and the potentiating effect of carbapenem bactericidal activity. Finally, blocking experiments identified Gnly as an important component of the antimicrobial activity of MAIT cell secretomes against *E. coli*. Altogether, these results demonstrate potent antimicrobial activity of MAIT cells against both cell-associated and free-living forms of *E. coli* and show that MAIT cells can act to restore the antibiotic effect of carbapenems against CREC.

MAIT cells promote antimicrobial effects through multiple mechanisms, including killing of infected cells, induction of nitric oxide production, and orchestrating downstream effector cell responses [14,25,34–39]. Here, our findings indicate that both Gnly and GrzB expression are required for efficient MAIT cell control of cell-associated bacteria, independent of infected cell death and production of pro-inflammatory cytokines. Consistent with this model, a recent

**A**    MAIT cell antimicrobial activity against cell-associated *E. coli*

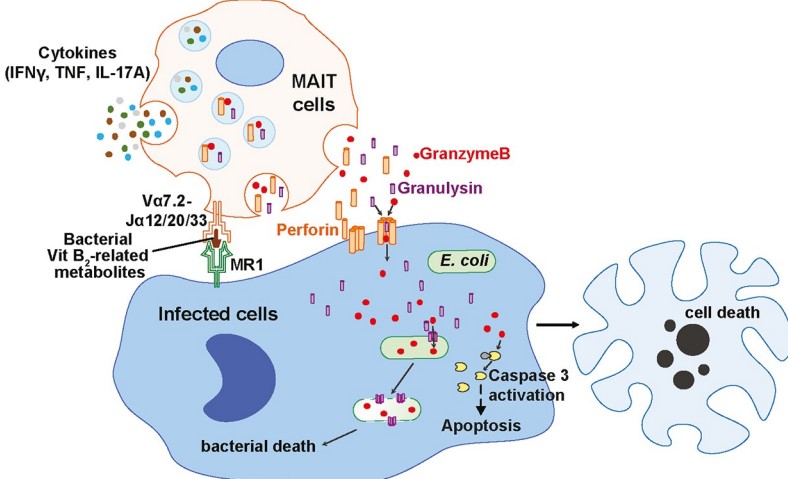

**B**    MAIT cell antimicrobial activity against extracellular *E. coli*

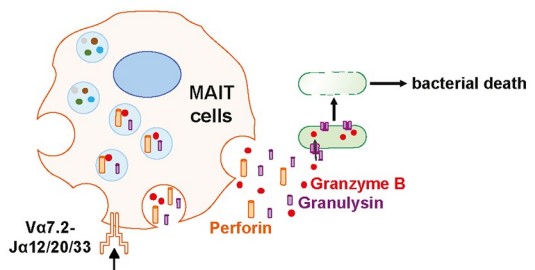

**C**    MAIT cell secretomes potentiation of carbapenems against carbapenem-resistant *E. coli*

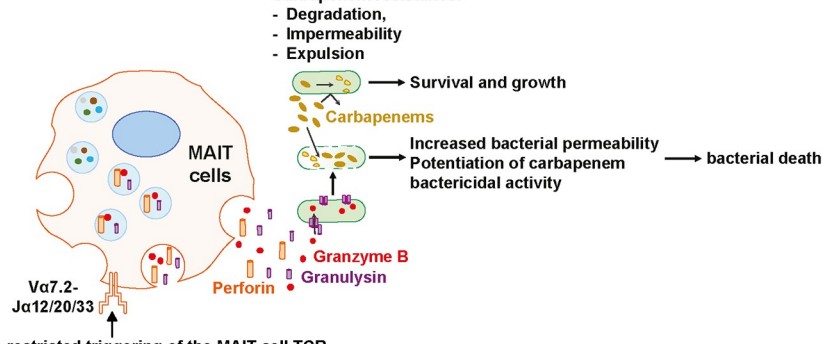

**Fig 6. A model of MAIT cell antimicrobial activity against cell-associated (A) and extracellular (B) drug-sensitive *E. coli* and CREC, and (C) MAIT cell secretome potentiation of carbapenem killing activity against carbapenem-resistant *E. coli* strains.** CREC, carbapenem-resistant *E. coli*; IFNγ, interferon-γ; IL-17A; interleukin-17A MAIT, Mucosa-associated invariant T; MR1, MHC-Ib-related protein; TCR, T cell receptor; TNF, tumor necrosis factor.

study revealed the existence of an adaptive CD8[+] T cell subset co-expressing Prf, GrzB, and Gnly, which can kill intracellular bacteria [40]. Our data further indicate that MAIT cells also control extracellular bacteria using the combined activity of Gnly and GrzB. MAIT cells secreted high levels of Gnly and GrzB into the surrounding milieu in response to MR1-restricted activation. Gnly is a cationic saposin-like antimicrobial protein that binds to and forms

pores on cholesterol-poor bacterial membranes and is widely known to kill bacteria by disrupting membrane permeability, including those of *E. coli* [32,41–43]. Consistent with this, we detected the presence of Gnly inside bacterial cells exposed to MAIT cell secretomes, and this was associated with increased bacterial permeability and damage. Additionally, we detected intracellular GrzB in bacterial cells incubated with MAIT cell secretomes, but only when Gnly was also present. The dual presence of GrzB and Gnly further increased bacterial permeability and damage. In the context of cytolytic protein secretion by cytotoxic CD8[+] T cells, Gnly-mediated pore formation in bacterial cell walls allows GrzB to penetrate into bacterial cytoplasm and cleave oxidative stress defense enzymes and other proteins vital for bacterial survival. This ultimately leads to the generation of reactive oxygen species (ROS) and bacterial death [29,30]. Interestingly, in a mouse model of *Legionella* infection, cytolytic proteins did not appear to contribute to the MAIT cell protective role [38]. However, because cytolytic proteins are structurally and functionally divergent in mammals [44], and Gnly is absent in rodents [45], prudence should be exercised in interpreting the mechanism underlying MAIT cell mediated antimicrobial activity in murine models. Collectively, our findings indicate that MAIT cells mediate antimicrobial activity against both cell-associated and extracellular forms of *E. coli* through the cytolytic protein-dependent pathway, which may contribute to the protection of the human host against bacterial infections.

An important aspect of the present study concerns the differential and temporal regulation of MAIT cell cytotoxicity and antimicrobial activity. Resting MAIT cells expressed GrzA, GrzK, and Prf but were negative for GrzB and Gnly, with no appreciable cytotoxic or antimicrobial capacity. Following antigenic stimulation, there was rapid and strong up-regulation of GrzB, accompanied by strong cytotoxicity but inefficient antimicrobial activity, and finally a gradual expression of Gnly coupled with strong cytotoxicity and antimicrobial activity. How long these cytolytic effector proteins remain expressed by MAIT cells following antigenic stimulation and resolution of bacterial infection, and whether such temporal regulation influences MAIT cells' protective roles of the host is unclear. We previously observed variable expression levels of Gnly in resting PB MAIT cells [2,25]. As Gnly is similarly expressed late by cytotoxic T cells following activation [46], Gnly[+] MAIT cells in blood may represent an antigen-experienced population that may respond to bacterial infection faster. Although Gnly is broadly antimicrobial, it also has strong pro-inflammatory and chemotactic activity and has been implicated in several inflammatory diseases [45]. Thus, the differential and temporal regulation of cytotoxicity and antimicrobial activity may also afford some protection against MAIT cell-mediated immunopathology that can occur in bacterial infections [47,48]. Interestingly, cytolytic proteins are differentially expressed during the development of human MAIT cells, with the mature subsets expressing more of these effector proteins [49]. In adult human female genital tract, oral mucosa, and placental tissues, MAIT cells express low level of GrzB at steady state [12,50–52], suggesting that the regulation of cytolytic effector proteins in tissues may differ from that of blood MAIT cells. Here, we observed that tissue-resident NP MAIT cells express somewhat lower levels of cytolytic proteins at baseline and following in vitro culture, although the acquisition of cytolytic proteins occurs at comparable rates. Furthermore, tissue-resident NP MAIT cells are capable of killing *E. coli*–infected cells. Given the abundance of MAIT cells in mucosal tissues, resident MAIT cells may thus play an important role in early bacterial control in the mucosal tissue environments.

CRE organisms, including CREC, have a high risk of transmission and rapid spread between patients in the hospital settings [20]. Carbapenems disrupt bacterial cell wall synthesis, and the resistance to carbapenems in Enterobacteriaceae often involves carbapenem impermeability, efflux pumps, and expression of carbapenemases. Combined with the increasing occurrence of pan-drug resistance, the spread of CRE organisms are a rising threat to

public health [20]. Few studies have considered the role of the host's immune response in facilitating clearance of resistant populations arising during the course of antimicrobial therapy. Yet, it is possible that the host's immune response during the course of antimicrobial chemotherapy may limit the appearance of resistant populations [53–55]. This may help explain why treatments with single antimicrobial agents are often successful in healthy, immunocompetent individuals [56]. In the present study, we show that MAIT cells recognize and respond to CREC-infected cells and mediate killing of infected cells and control of bacterial loads. Our findings that MAIT cells retain their ability to mediate antimicrobial activity against resistant bacteria implies that MAIT cells play a role in defense not only against drug-sensitive bacteria but also against drug-resistant bacteria. Consistent with this hypothesis, individuals with comorbidities that are frequently linked with low MAIT cell numbers and poor antimicrobial functionality [57–60] are more susceptible to infections caused by drug-resistant bacterial pathogens [61–64]. This supports a potential importance of the MAIT cell antimicrobial effector response in the defense against drug-resistant bacterial pathogens. It is thus tempting to speculate that CRE infections among at-risk individuals may be preventable by restoring MAIT cell numbers and functionality through vaccinations using MR1 ligands [39].

Finally, and strikingly, high levels of cytolytic effector proteins secreted by MAIT cells into the surrounding milieu potentiate the bactericidal activity of carbapenems against extracellular CREC primary clinical isolates. The enhancement of carbapenem activity in vitro is likely due to increased bacterial membrane permeability and bacterial damage mediated by high levels of both Gnly and GrzB, allowing entry of lethal amounts of carbapenems into the bacteria. Thus, it is tempting to hypothesize that Gnly- and GrzB-dependent MAIT cell secretome antimicrobial activity is not only able to control bacterial growth but also able to disarm carbapenem impermeability mechanisms of resistance in free-living CREC by increasing carbapenem access to bacterial intracellular space. In some circumstances, however, the biological activity of Gnly and GrzB in our static culture system appeared to be relatively limited, because in rare experiments the CREC strains rebounded and grew back even in the presence of imipenem and MAIT cell secretome. This limitation was strongly reduced when lower starting inocula were used, consistent with the notion that there was a finite amount of biologically active Gnly and GrzB in the MAIT cell secretomes in our static culture system. Future studies to investigate the precise mechanism of how Gnly and GrzB overcome carbapenem resistance in CREC strains in vitro should therefore use a dynamic culture system in which the biological activity and properties of secreted GrzB and Gnly could be simulated.

In summary, our findings define the mechanism by which MAIT cells mediate direct antimicrobial activity and exert early bacterial control in vitro. The finding that MAIT cells use Gnly and GrzB activity to kill both extracellular and cell-associated *E. coli* suggests that it may be difficult for bacteria to develop resistance to MAIT cell antimicrobial activity because of the conserved targeting of vital survival pathways by Gnly and GrzB. Going forward, this may be further explored in vivo using transgenic murine models, which will allow dissection of the requirements of individual cytolytic proteins for human MAIT cell cytotoxic capacity and early defense against multidrug-resistant bacteria. In conclusion, the present study indicates that MAIT cell cytolytic properties are an important component of their antimicrobial effector function and enable enhancement of carbapenem killing activity against carbapenem-resistant *E. coli* strains. The ability of MAIT cell antimicrobial effector function to overcome drug resistance suggests that MAIT cells may participate in clearance of bacteria acquiring de novo resistance during the course of antibiotic therapy.

## Materials and methods

### Ethics statement

Human samples were collected after informed written consent was obtained from all donors in accordance with study protocols conforming to the provisions of the Declaration of Helsinki. Ethical approval was obtained from the National University of Singapore Institutional Review Board (NUS-IRB reference codes B-15-088 and H-18-029) and Singapore General Hospital Institutional Review Board (protocol 2019/2623).

### Blood and tissue processing

PB was collected from healthy donors recruited at the apheresis unit, Bloodbank@HSA, Health Services Authority, Singapore. PBMCs were isolated by standard Ficoll-Histopaque density gradient separation (Ficoll-Histopaque Premium; GE Healthcare). After isolation, PBMCs were cryopreserved in liquid nitrogen until further use or immediately used for MAIT cell purification.

Anonymized matched blood and NP tissue samples were collected from healthy individuals from Singapore General Hospital; Singapore. PBMCs were isolated by density gradient centrifugation on Lymphoprep (Axis Shield). PBMCs and NP tissue biopsies were kept frozen in 10% DMSO (Sigma-Aldrich), 90% Fetal Bovine Serum (Hyclone). Before initiation of tissue-resident MAIT cell culture, frozen tissue biopsies were thawed, washed, and disinfected with 5× antibiotic-antimycotic solution (Gibco) supplemented-RPMI-1640 (Hyclone). Tissues were digested using human tumor dissociation kit (Miltenyi Biotec), then passed through nylon mesh to obtain single-cell suspensions.

### MAIT cell expansion

MAIT cells were expanded using 2 different protocols in this study. For the first set of expansion, MAIT cell were isolated from freshly isolated PBMCs by positive selection using the 5-OP-RU-loaded human MR1 tetramer-PE using magnetic-activated cell sorting (MACS) and anti-PE microbeads (Miltenyi). MAIT cells (>95% purity) were cultured 6 to 7 d in serum-free and xeno-free ImmunoCult-XF T cell expansion medium (STEMCELL Technologies) in the presence of 1:100 ImmunoCult human CD3/CD28/CD2 polyclonal T cell activator (STEMCELL Technologies), 10 ng/mL recombinant human (rh)IL-7 (R&D Systems), 50 μg/mL gentamicin (Gibco), and 100 μg/mL normocin (Invivogen). In selected experiments, MAIT cells were purified using the anti-Vα7.2 beads as described [24], then either immediately used or cultured for 2 or 15 d in the presence of 5 ng/mL rhIL-2 (Peprotech) and 10 ng/mL rhIL-7 before use.

For the second set of expansion, cryopreserved PBMCs were thawed and cultured in ImmunoCult-XF T cell expansion medium supplemented with 8% (v/v) xeno-free CTS Immune Cell Serum Replacement (ThermoFisher Scientific), 5 ng/mL rhIL-2 (Peprotech), 10 ng/mL rhIL-7, 50 μg/mL gentamicin, and 100 μg/mL normocin. PBMCs were stimulated with 10 nM 5-OP-RU on day 0, 5, and 10, and the culture media was replenished every 2 to 3 d. On day 11, viable cells were isolated by Ficoll-Histopaque density gradient centrifugation. On day 15 through 17, cells were checked for MAIT cell purity and numbers by flow cytometry. Cells were immediately used for subsequent assays when MAIT cell purity >70%. Unless otherwise indicated, all MAIT cell in vitro expansion used the second set of expansion protocol.

### Bacterial cultures

The *E. coli* strains 1100–2 and BSV18 were obtained from the Coli Genetic Stock Center, Yale University; the DH5α strain was obtained from New England Biolab. Carbapenem-resistant

isolates were identified from the Singapore General Hospital microbiological database and retrieved from the archived bacteria repository. Further details on strain identification and determination of resistance and susceptibility profiles can be found in S1 and S2 Tables and S1 Text.

For MAIT cell stimulation assays, all *E. coli* strains were grown overnight at 37 ˚C in Luria (lysogeny) broth (LB) with shaking as described [65]. Overnight cultures of *E. coli* were then subcultured 10-fold in LB and incubated at 37 ˚C with shaking until $OD_{600}$ = 0.5. In selected experiments, growth curves were monitored by reading absorbance at 600 nm in a microplate reader with discontinuous shaking for 18 h at 37 ˚C (Cytation 5, BioTek Instruments).

## Preparation of MAIT cell antigen 5-OP-RU

The MAIT cell antigen 5-OP-RU was synthesised according to a previously published procedure [31]. It is stable in DMSO solutions but converts rapidly to much less active lumazine in aqueous media, the exposure time to which should be minimized as much as possible to maximize activity. All 5-OP-RU working solutions were diluted from the DMSO stocks with appropriate culture medium immediately prior to any functional assay.

## MAIT cell functional and antimicrobial activity assays

MAIT cells within bulk PBMCs, identified as MR1-5-OP-RU$^+$ Vα7.2$^+$ CD161$^{hi}$ CD3$^+$ T cells, were activated using formaldehyde-fixed *E. coli* strains as indicated for 24 h as previously described [24]. In selected experiments, 20 μg/mL MR1 blocking mAb (26.5, Biolegend) or IgG2a isotype control (MOPC-173, Biolegend) were used [24]. In all activation assays, monensin (Golgi Stop, BD Biosciences) were added for the last 6 h of incubation.

MAIT cell cytotoxicity assay was performed as previously described [24,25]. Briefly, HeLa cells or human 293T cells stably transfected with human MR1 (293T-hMR1) as indicated were incubated in complete RPMI medium with formaldehyde-fixed *E. coli* at the microbial dose of 30 for 3 h before the addition of expanded MAIT cells at the effector to target cell ratio 5:1. A total of 2 nM 5-OP-RU was added as a positive control for MAIT cell killing of target cells. Anti-CD107a-BUV395 at 1:200 was added at the beginning of the assay to detect MAIT cell degranulation. After 24 h of co-culture, cells were stained to detect target cell apoptosis using anti-active Casp3 mAb (BD Biosciences).

MAIT cell antimicrobial activity assay was performed as described [66]. Briefly, the bacteria subcultures were resuspended in RPMI without serum and antibiotics (ASF-RPMI). Adherent target epithelial cell lines were infected with live *E. coli* for 3 h at a microbial dose of 30 for strain EC120S and 3 for strain EC241 at 37 ˚C / 5% $CO_2$. Infected cells were washed extensively with complete RPMI medium supplemented with 200 μg/mL gentamicin (Gibco) and further incubated for 1 h at 37 ˚C / 5% $CO_2$ to kill extracellular bacteria, then washed extensively with ASF-RPMI. Expanded MAIT cells were labeled with 1 μM CellTrace Violet (CTV) dye (Thermo Fisher Scientific) before co-cultured with HeLa cells at an E:T ratio of 5:1 and incubated for 3 h at 37 ˚C / 5% $CO_2$. For the live bacteria enumeration, a duplicate set of experimental wells were done in parallel. Supernatants from the first set were collected, followed by adherent cell lysis with 0.1% (v/v) Triton-X for 10 min at RT. Equivalent volume of LB broth were added to lysates and plated onto LB agar plates in duplicates, incubated at 37 ˚C for 18 to 24 h, and counted visually. For the second set, anti-CD107a-BUV395 (1:200) was added into the culture medium at the start of the assay to assess MAIT cell degranulation. Cell-free supernatants were harvested, snap frozen in liquid nitrogen, and stored at −80 ˚C until further use. Adherent cells were harvested using trypsin-EDTA (Gibco) and stained for flow cytometry as indicated.

In selected experiments, MAIT cells or infected HeLa cells were treated with various pharmacological inhibitors or mAbs before use in assays. Briefly, MAIT cells were pre-incubated for 1 h with 5 mM EGTA (Bioworld) supplemented with 1 mM $MgCl_2$ (Sigma-Aldrich), then diluted to 1 mM EGTA with 1 mM $MgCl_2$ in-assay, or with 10 μM nafamostat mesylate (Sigma-Aldrich), or 100 μM of the GrzB inhibitor II Ac-IETD-CHO (Merck) before co-culture with HeLa cells. HeLa cells were pretreated for 1 h before addition of MAIT cells with 10 μM CAS-BIND Pro Pan Caspase Inhibitor (Pro-VAD-FMK; Vergent Bioscience). For antibody-blocking experiments, 20 μg/mL anti-MR1 or IgG2a isotype control was added on HeLa cells 1 h prior to co-culture, or 10 μg/mL mAb to IFNγ (B27; Biolegend), TNF (MAb1; Biolegend), IL-17A (eBio64CAP17; Invitrogen) or IgG1 isotype control (MOPC-21; Biolegend) 15 min before co-culture.

### GrzB activity detection

GrzB activity was measured with GranToxiLux PLUS! (GTL) Kit (Oncoimmunin, Inc). Briefly, HeLa cells were trypsinized and infected with *E. coli* as per MAIT cell antimicrobial assay and MAIT cells were co-cultured with HeLa cells at E:T ratio of 5:1. Co-cultured cells were centrifuged immediately and resuspended in GrzB substrate solution. The co-culture was done for 1 h at 37 °C, and cells were washed in the GTL Wash Buffer. Cells were stained with viability dye and immediately analyzed by flow cytometry.

### Preparation of MAIT cell supernatants

293T-hMR1 cells were plated in a 96-well flat bottom plate at 37 °C / 5% $CO_2$ in complete RPMI medium. After overnight incubation, culture medium was replaced with the antibiotic-free ImmunoCult medium and 2 nM 5-OP-RU was added into each well for 2 h. Expanded MAIT cells were then further purified using the Vα7.2 beads as described [24] (MAIT cell purity >98%) and co-cultured with the 5-OP-RU-pulsed 293T-hMR1 cells at an E:T ratio of 10:1. Supernatants from the co-culture wells or 5-OP-RU-pulsed 293T-hMR1 cell control wells were collected 24 h after co-culture, clarified, and snap frozen in liquid nitrogen until further use. The cytokine and cytolytic protein contents of the supernatants were measured using the LEGENDplex human CD8/NK cell panel (Biolegend) as described [2].

### MAIT cell secretome antimicrobial activity assay

Overnight bacteria subcultures were washed with PBS and resuspended in MAIT cell or control supernatants at $10^5$ CFU/mL and incubated at 37 °C in flat-bottom 96-well plates without shaking for 24 h. To enumerate live bacteria, bacterial suspensions were harvested at various time points as indicated, serially diluted in LB broth, plated in triplicates on LB agar plates, and incubated at 37 °C for 18 to 24 h and counted visually. In selected experiments, MAIT cell and control supernatants were spiked with imipenem monohydrate (Sigma-Aldrich) at various concentrations as indicated. A 3 $log_{10}$ reduction in bacterial counts from the baseline population over 24 h was considered as bactericidal.

To assess bacterial damage, the cell-impermeable SYTOX Green nucleic acid stain (Thermo Fisher Scientific) was added at 20 μM during the 2 h incubation of $10^6$ CFU/mL *E. coli* with MAIT cell or control supernatants, with or without carbapenems at the indicated concentrations. Dead bacteria controls were prepared by incubating the bacteria in 70% (v/v) ethanol for 30 min at RT, washed, then incubated with control medium supplemented with SYTOX Green. To allow identification of the bacteria versus debris by flow cytometry, the bacteria were stained with SYTO 62 red fluorescent nucleic acid stain (Thermo Fisher Scientific; 2.5 μM) during the last 15 min at 37 °C. Bacteria were then fixed with 1% formaldehyde for 20

min at 4 ˚C just prior to FACS acquisition. In selected experiments, to detect the intracellular cytolytic proteins in bacterial cells, $10^5$ CFU/mL of *E. coli* were cultured with control or MAIT cell supernatants in the presence of 20 μM SYTOX Green for 30 min at 37 ˚C without shaking. During the last 15 min of incubation, fluorochrome-conjugated mAbs against Gnly, GrzA, GrzB, and Prf (S3 Table), as well as SYTO 62 nucleic acid dye were added to the culture. Bacteria were then washed, fixed, and permeabilized with BD Cytofix/Cytoperm buffers (BD Biosciences), then restained with the same mAb cocktail for 30 min at 4 ˚C and washed once prior to FACS acquisition.

## Flow cytometry analysis

Cell surface and intracellular staining for cytokines, cytotoxic molecules, and active Casp3 were performed as previously described [2]. Staining with the MR1 5-OP-RU and MR1 6-FP tetramers was performed for 40 min at room temperature (RT) [7] before proceeding to the surface and intracellular staining with other mAbs (S3 Table.) Samples were acquired on an LSRFortessa flow cytometer (BD Biosciences) equipped with 355, 405, 488, 561, and 640 nm lasers. Single-stained polystyrene beads (BD Biosciences) and the compensation platform in FACSDiva version 8.0.1 (BD Biosciences) or FlowJo software versions 9.9 and 10.5 (TreeStar) were used for compensation.

## Statistical analysis

Statistical analyses were performed using Prism software version 8.3.0 (GraphPad). Data sets were first assessed for data normality distribution. Data presented as a heat map shows the mean, whereas data presented as line or bar graphs with error bars represent the mean and standard error. Box and whisker plots show median, the 10th to 90th percentile, and the interquartile range. Statistically significant differences between samples were determined as appropriate using the unpaired *t* test or Mann-Whitney's test for unpaired samples, and the paired *t* test or Wilcoxon's signed-rank test for paired samples. The Kruskal-Wallis one-way ANOVA, the Friedman test, ordinary ANOVA, the repeated-measures (RM) one-way ANOVA, or mixed-effects analysis followed by the appropriate post hoc test as indicated was used to detect differences across multiple samples. Correlations were assessed using the Pearson correlation or Spearman rank correlation for parametric or nonparametric data, respectively. Two-sided $p < 0.05$ were considered significant.

## Supporting information

**S1 Data. Raw data for all main and supplemental figures.**
(XLSX)

**S1 Text. Additional materials and methods.**
(DOCX)

**S1 Table. The *E. coli* clinical isolates used in this study.**
(DOCX)

**S2 Table. The MIC of the *E. coli* clinical isolates.** MIC, minimum inhibitory concentrations.
(DOCX)

**S3 Table. Flow cytometry-based antibodies and reagents used in the study.**
(DOCX)

**S1 Fig. MAIT cells killed *E. coli*–infected cells and suppressed bacterial loads.** (A) An illustration of the protocol for the killing assay of *E. coli*–infected HeLa cells by MAIT cells and MAIT cell antimicrobial activity, along with a representative FACS plot of MAIT cell proliferation on day 6 using CTV dilution assay and % MAIT cells (left middle panels) and expression of Gnly and GrzB (left bottom panels) following 14 d of in vitro expansion, and (B) the gating strategy for the flow cytometry analyses. (C) Representative fluorescence microscopy images of MAIT cells (blue)-mediated killing of HeLa cells (green) following internalization of pHrodo red-labeled live *E. coli* EC120S (red) by HeLa cells depicted by white arrows (*n* = 3 independent experiments). (D) pHRodo-labeled *E. coli* strain EC120S uptake by HeLa and A549 cells for 3 h on ice or at 37 ˚C (*n* = 3 independent experiments). (E) Bacterial loads of *E. coli* strain EC120S over 24 h at 37 ˚C inside the infected HeLa cells and in supernatants in the presence of low concentration (20 µg/mL) of gentamicin (*n* = 4–11 per time point). (F) Representative flow cytometry staining of active Casp3 and amine-reactive dead cell marker in A549 cells alone, A549 cells infected with *E. coli* EC120S, or A549 cells co-cultured with MAIT cells with or without *E. coli* EC120S for 24 h. (G) Representative flow cytometry plot of CD107a/degranulation in MAIT cells alone, or co-cultured with A549 cells with or without *E. coli* EC120S. (H) Bacterial counts in *E. coli* EC120S-infected A549 cells co-cultured with or without MAIT cells for 24 h (*n* = 4). (I, J, K) Apoptosis of HeLa cells (I), degranulation of effector cells (J), and bacterial counts (K) in the HeLa-MAIT or HeLa-Vα7.2⁻ T cells co-culture with or without *E. coli* EC120S (*n* = 5–6 in panels I and K and *n* = 8 in panel J). Data presented as line with error bars represent the mean and standard error. Box and whisker plots show median, the 10th to 90th percentile, and the interquartile range. Statistical significance was determined using mixed-effects analysis followed by Tukey's post hoc test (E), the Mann-Whitney test (I), Wilcoxon's signed-rank test (J), and the Friedman multiple comparisons test followed by Dunn's post hoc test (K). $^{**}p < 0.01$, $^{*}p < 0.05$, [*]$p < 0.1$. The underlying data of this figure can be found in S1 Data. Casp, caspase; CFU; colony-forming units; CTV, CellTrace Violet; DCM, dead cell marker; FACS, fluorescence-activated cell sorting; FAM, fluorescein amidite; FLICA, fluorescence inhibitor of caspase activation; FSC-A, forward-scatter area; Gnly, granulysin; Grz, Granzyme; MAIT, Mucosa-associated invariant T; ns, not significant; SSC-A, side-scatter area. (TIF)

**S2 Fig. Expression of cytolytic proteins in MAIT cells is temporally regulated.** (A) Representative flow cytometry staining of Casp3 expression in HeLa cells and CD107a degranulation in MAIT cells stimulated with *E. coli* EC120S for 24 h using MAIT cells from D0, D2, and D15 after expansion (protocol 2). (B, C) Casp3 expression in HeLa cells and CD107a degranulation in MAIT cells stimulated with the MR1 ligand 5-OP-RU for 24 h using MAIT cells from D0 and D2 and D15 after expansion (all *n* = 4). (D, E) Representative flow cytometry data (D) and combined data (E) of GrzA, GrzB, GrzK, Gnly, and Prf (*n* = 4–10) levels (MFI) in MAIT cells over the course of the in vitro expansion. (F) Identification of matched PB and tissue-resident MAIT cells from the NP mucosae of 3 healthy individuals undergoing nasal polyp removal. (G) Relative expression levels (fold change of MFI to D0) of cytolytic proteins expressed by matched PB and NP MAIT cells at baseline and at various time points following in vitro expansion (*n* = 3–4). (H) Detection of cytolytic protein contents in the effector MAIT cells and target *E. coli* EC120S-infected HeLa cells following 3 h co-culture with MAIT cells in the presence or absence of EGTA + Mg²⁺. Representative histograms from at least 2 independent MAIT cell donors are shown. (I) Levels of cytokines in the supernatants following MAIT cell co-culture with *E. coli* EC120S-infected HeLa cells for 3 h (*n* = 6). Data presented as line or bar graphs with error bars represent the mean and standard error. Box and whisker plots show median, the 10th to 90th percentile, and the interquartile range. The underlying data of this

figure can be found in S1 Data. Casp, caspase; D, day; Gnly, granulysin; Grz, Granzyme; IFNγ, interferon-γ; IL-17A, interleukin-17A; MAIT, Mucosa-associated invariant T; MFI, mean fluorescence intensity; MR1, MHC-Ib-related protein; NP, nasopharyngeal; PB, peripheral blood; Prf, perforin; 5-OP-RU, 5-(2-oxopropylideneamino)-6-D-ribitylaminouracil.
(TIF)

**S3 Fig. MAIT cells responses to stimulation with CREC clinical strains.** (A–H) Growth curve of the *E. coli* strains BSV18 (RibA⁻), 1100–2 (RibA⁺ isogenic strain of BSV18), EC120S, EC234, EC241, EC362, and EC385 in LB or in riboflavin-deficient medium with supplemental riboflavin or acetonitrile solvent control (*n* = 3). (I) Relative RNA expression of *RibA* of the indicated *E. coli* (*n* = 3 independent experiments). (J) Representative flow cytometry plots of degranulation (CD107a) and production of GrzB, IFNγ, TNF, and IL-17A by MAIT cells following stimulation of PBMCs with formaldehyde-fixed *E. coli* strains DH5α, EC120S, EC234, and EC362. (K) Polyfunctional profile of MAIT cell responses against the indicated *E. coli* strains presented in pie charts (*n* > 5). Comparison of the pie chart distributions was performed using a partial permutation test and performed using SPICE version 5.1, downloaded from http://exon.niaid.nih.gov [6] (L) Bacterial uptake by PBMC (*n* = 3) in the presence of pHrodo-labeled *E. coli* strains as indicated for 3 h on ice or at 37 ˚C. (M) Representative flow cytometry plots of Casp3 activation and apoptosis in 293T-hMR1 cells alone, 293T-hMR1 cells infected with EC234, or co-culture with MAIT cells with or without EC234 for 24 h. (N, O) Casp3 activation and apoptosis in 293T-hMR1 cells alone or co-cultured with MAIT cells in the presence of 5-OP-RU or *E. coli* strains DH5α, EC120S, EC234, or EC362 (*n* = 7–9). MAIT cells were expanded polyclonally for 7 d as described. (P) Bacterial uptake by HeLa cells (*n* = 3 independent experiments) in the presence of pHrodo-labeled *E. coli* EC241 for 3 h on ice or at 37 ˚C. Data presented as line graphs represent the mean, and bar graphs with error bars represent the mean and standard error. Box and whisker plots show median, the 10th to 90th percentile, and the interquartile range. Statistical significance was determined using the Friedman test (N, O) followed by Dunn's multiple comparison test. **$p < 0.01$, *$p < 0.05$. The underlying data of this figure can be found in S1 Data. Abs, absorbance; AU; arbitrary unit; Casp, caspase; CREC, carbapenem-resistant *E. coli*; DCM, dead cell marker; Grz, Granzyme; IFNγ, interferon-γ; IL-17A, interleukin-17A; LB, Luria (lysogeny) broth; MAIT, Mucosa-associated invariant T; MR1, MHC-Ib-related protein; ns, not significant; PBMC, peripheral blood mononuclear cell; RAM; riboflavin assay media; *RibA*; gene encoding GTP cyclohydrolase-2; SSC-A; side-scatter area; TNF, tumor necrosis factor; 293T-hMR1, 293T cells stably transfected with human MR1; 5-OP-RU, 5-(2-oxopropylideneamino)-6-D-ribitylaminouracil.
(TIF)

**S4 Fig. Antimicrobial activity of MAIT cell secretomes.** (A) An illustration of the protocol to test the antimicrobial activity of the MAIT cell secretomes using flow cytometry stainings of the bacteria and live bacterial counts using the traditional agar plating method. (B) Representative flow cytometry plots of MAIT cell degranulation (CD107a) and expression of Gnly and GrzB following stimulation with 5-OP-RU-pulsed 293T-hMR1 cells. (C) Concentration of Gnly, GrzA, GrzB, and Prf secreted by MAIT cells and Va7.2⁻ T cells stimulated with 5-OP-RU-pulsed 293T-hMR1 cells. (D) Representative flow cytometry plots of SYTOX Green expression in *E. coli* strain EC234 untreated (live) or killed with 70% (v/v) ethanol. (E, F) Representative flow cytometry histograms (E) and combined data (F) of SYTO 62 expression in *E. coli* strains EC234 and EC362 in the presence of MAIT cell or control supernatants (*n* = 6). (G) Live *E. coli* EC362 bacterial counts over 24 h in the presence of control or supernatants prepared from MAIT cells expanded polyclonally for 7 d as described (*n* = 4–6). (H) SYTOX Green MFI in EC234 and EC362 in the presence of MAIT cell or control supernatants and

with or without imipenem for 2 h ($n = 8–10$). (I, J, K) Representative flow cytometry plots of SYTOX Green staining (I), and histograms (J) and combined data (K) of SYTO 62 staining and levels on strains EC234 and EC362 following 2 h incubation with MAIT cell or control supernatants with or without imipenem ($n = 6–8$), ertapenem ($n = 3$ for both strains), or meropenem ($n = 3$ for both strains). Statistical significance was calculated using the paired *t* test (F), two-way ANOVA or mixed-effects analysis with Sidak's multiple comparison test (G), and RM one-way ANOVA with Dunnett's multiple comparisons test (H, J). The bar graphs and error bars represent the mean and standard error, whereas scatter plots show median and the IQR. ****$p < 0.0001$, ***$p < 0.001$, **$p < 0.01$, *$p < 0.05$, [*]$p < 0.1$. The underlying data of this figure can be found in S1 Data. CFU, colony-forming units; Ctrl, control; EtOH; ethanol; Gnly, granulysin; Grz, Granzyme; IQR, interquartile range; MAIT, Mucosa-associated invariant T; MFI, mean fluorescence intensity; PBMC, peripheral blood mononuclear cell; Prf, perforin; s/n; supernatant; 239T-hMR1, 293T cells stably transfected with human MR1; 5-OP-RU, 5-(2-oxopropylideneamino)-6-D-ribitylaminouracil.
(TIF)

**S5 Fig. MAIT cell secretomes restore imipenem bactericidal activity against CREC strains.** (A, B) Growth curves (A) and the lag phase (B) of the *E. coli* strains EC234 and EC362 in presence of MAIT cell or control supernatants and at the indicated concentration of imipenem ($n = 2–14$, both strains). (C) Heat map of detectable growth of strains EC234 and EC362 after 18 h incubation in MAIT cell or control supernatants and at the indicated concentration of imipenem ($n = 10$ [EC234], 11 [EC362]). (D) Relative bacterial loads of strains EC234 and EC362 in presence of supernatants derived from 5-OP-RU-pulsed 293T cells alone (control), or following co-culture with MAIT cells or V$\alpha$7.2$^-$ T cells. (E, F) Representative flow cytometry plots of degranulation (CD107a) (E) by V$\alpha$7.2-bead-purified and -activated MAIT cells without further co-culture with Ag-presenting cells and that of bulk MAIT cells co-cultured with 5-OP-RU-pulsed 293T-hMR1 cells. *E. coli* strain EC362 was then incubated with clarified supernatants that were collected 24 h after co-culture in the presence of imipenem (F) and the bacterial loads were determined over time ($n = 2$). (G) Representative flow cytometry plots of degranulation by MAIT cells and Casp3 activation in 293T-hMR1 cells in the co-culture untreated or in the presence of 5-OP-RU with or without the pan-Casp inhbitor Pro-VAD. (H) Relative bacterial loads of the strains EC234 or EC362 after 24 h incubation with imipenem-supplemented MAIT cell supernatants prepared in the presence or absence of Pro-VAD ($n = 4$ [EC234], 6 [EC362]). Significant differences between control and MAIT cell supernatants at indicated imipenem concentrations were calculated using mixed-effects analysis with Sidak's multiple comparisons test (A–C). The heat map shows the mean, the lines of the growth lag-phase curves represent the mean, the bar graphs and error bars represent the mean and standard error, whereas box and whisker plots show median, the 10th to 90th percentile, and the IQR. The underlying data of this figure can be found in S1 Data. Abs, absorbance; AU, arbitrary unit; Casp, caspase; CFU, colony-forming units; Ctrl, control; CREC, carbapenem-resistant *E. coli*; IQR, interquartile range; MAIT, Mucosa-associated invariant T; Pro-VAD, valyl-alanyl-aspartyl-[O-methyl]-fluoromethylketone; s/n, supernatant; 293T-hMR1, 293T cells stably transfected with human MR1; 5-OP-RU, 5-(2-oxopropylideneamino)-6-D-ribitylaminouracil.
(TIF)

**S6 Fig. Relationship between MAIT cell-derived cytolytic proteins and antimicrobial activity of MAIT cell secretomes.** (A–F) Correlation between the concentration of GrzA (A, B), Prf (C, D) and Gnly (E, F) in the supernatant of MAIT cells stimulated with 5-OP-RU-pulsed 293T-hMR1 cells with the bacterial loads of *E. coli* EC234 (A, C) and EC362 (B, D), or with the

lag-phase of strains EC234 (E) and EC362 (F) in the presence of the indicated concentration of imipenem ($n = 12$ in panels A and C; $n = 17$ in panel B; $n = 28$ in panel D; $n = 9$ in panel E; $n = 14$ in panel F). (G) Analyses of live bacterial counts of *E. coli* EC234 and EC362 based on the Gnly concentration levels following 24 h incubation in control or MAIT cell supernatants supplemented with imipenem. (H) Representative flow cytometry plots of MAIT cell degranulation following co-culture with 5-OP-RU-pulsed 293T-hMR1 cells in the presence or absence of EGTA, anti-IFNγ, anti-TNF, anti-IL-17A, or IgG1 isotype control. (I) The relative *E. coli* EC362 bacterial loads following 24 h incubation in MAIT cell supernatants to those of control supernatants (both supplemented with 2 μg/mL imipenem) spiked with the degranulation inhibitor EGTA, GrzB inhibitor IETD-CHO, and mAbs against IFNγ, -TNF, or -IL-17A. For EGTA and mAb/isotype control-treated supernatants, EGTA and respective mAbs were added during the 24 h MAIT cells + 293T-hMR1 cells + 5-OP-RU co-culture stage, then clarified supernatants were used for the antimicrobial activity assay as described. (J) Proportion of Gnly and other effector molecules remaining when compared to mock-depleted (IgG1 isotype-treated) supernatant following Gnly-specific depletion of MAIT cell supernatants ($n = 3–4$). (K) Growth curves of strain EC362 with 2 μg/mL of imipenem in the presence of control or MAIT cell supernatant, or in mock-, Gnly-, or Prf-depleted MAIT cell supernatants ($n = 3$). (L) SYTOX Green and SYTO 62 staining intensity (MFI) of strain EC362 following 2 h-treatment with 2 μg/mL of imipenem in the presence of mock- or Gnly-depleted MAIT cell supernatant ($n = 5$). Significant differences were calculated using the mixed-effects analysis with Tukey's (G) or Dunnett's multiple comparisons test (I), two-way ANOVA with Dunnett's multiple comparison test (K), or the Wilcoxon test (L). Correlations were determined using the Spearman (E) or the Pearson (F) correlation test. The lines of the growth curves represent the mean, and the line or bar graphs with error bars represent the mean and standard error. $****p < 0.0001$, $***p < 0.001$, $**p < 0.01$, $*p < 0.05$, $[*]p < 0.1$. The underlying data of this figure can be found in S1 Data. Abs, absorbance; AU, arbitrary unit; CFU, colony-forming units; EGTA, ethylene glycol tetraacetic acid; Gnly, granulysin; Grz, Granzyme; IETD-CHO, N-acetyl-L-isoleucyl-L-α-glutamyl-N-[(1S)-2-carboxy-1-formylethyl]-L-threoninamide trifluoroacetate; IFNγ, interferon-γ; IgG1, immunoglobulin G subclass 1; IL-17A, interleukin-17A; mAbs, monoclonal antibodies; MAIT, Mucosa-associated invariant T; MFI, mean fluorescence intensity; ns, not significant; Prf, perforin; s/n, supernatant; TNF, tumor necrosis factor; 293T-hMR1, 293T cells stably transfected with human MR1; 5-OP-RU, 5-(2-oxopropylideneamino)-6-D-ribitylaminouracil.
(TIF)

**S7 Fig. Detection of cytolytic proteins inside bacterial cells.** (A–C) Representative flow cytometry plots (A) and histograms (B) of cytolytic protein expression and (C) SYTOX Green staining intensity of *E. coli* strain EC362 treated with growth medium (LB), 70% ethanol, 293T-hMR1+5-OP-RU (control) supernatant, and 293T-hMR1+MAIT+5-OP-RU (MAIT) supernatant in the absence or presence of 2 μg/mL imipenem as indicated ($n = 6$). Ctrl, control; EtOH, ethanol; Gnly; granulysin; Grz, granzyme; LB, Luria (lysogeny) broth; MAIT, Mucosa-associated invariant T; Prf, perforin; s/n, supernatant; 293T-hMR1, 293T cells stably transfected with human MR1; 5-OP-RU, 5-(2-oxopropylideneamino)-6-D-ribitylaminouracil.
(TIF)

## Acknowledgments

We thank Dr. Ted Hansen for the kind gift of 293T-hMR1 cell line. The MR1 tetramer technology was developed jointly by Dr. James McCluskey, Dr. Jamie Rossjohn, and Dr. David

Fairlie; and the material was produced by the NIH Tetramer Core Facility as permitted to be distributed by the University of Melbourne.

## Author Contributions

**Conceptualization:** Johan K. Sandberg, Edwin Leeansyah.

**Data curation:** Caroline Boulouis, Wan Rong Sia, Muhammad Yaaseen Gulam, Jocelyn Qi Min Teo, Edwin Leeansyah.

**Formal analysis:** Caroline Boulouis, Wan Rong Sia, Edwin Leeansyah.

**Funding acquisition:** Andrea Lay Hoon Kwa, Johan K. Sandberg, Edwin Leeansyah.

**Investigation:** Caroline Boulouis, Wan Rong Sia, Muhammad Yaaseen Gulam, Yi Tian Png, Thanh Kha Phan, Chwee Ming Lim, Edwin Leeansyah.

**Methodology:** Caroline Boulouis, Wan Rong Sia, Muhammad Yaaseen Gulam, Jocelyn Qi Min Teo, Yi Tian Png, Thanh Kha Phan, Chwee Ming Lim.

**Project administration:** Andrea Lay Hoon Kwa, Johan K. Sandberg, Edwin Leeansyah.

**Resources:** Jocelyn Qi Min Teo, Jeffrey Y. W. Mak, David P. Fairlie, Ivan K. H. Poon, Tse Hsien Koh, Peter Bergman, Chwee Ming Lim, Lin-Fa Wang, Andrea Lay Hoon Kwa, Johan K. Sandberg.

**Supervision:** Johan K. Sandberg, Edwin Leeansyah.

**Validation:** Wan Rong Sia, Muhammad Yaaseen Gulam.

**Writing – original draft:** Caroline Boulouis, Wan Rong Sia, Johan K. Sandberg, Edwin Leeansyah.

**Writing – review & editing:** Caroline Boulouis, Johan K. Sandberg, Edwin Leeansyah.

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
