## [Editor Report · Decision Letter 0]

9 Jan 2020

Dear Dr Leeansyah, 

Thank you for submitting your manuscript entitled "Human MAIT cell cytolytic effector proteins synergize to overcome carbapenem resistance in Escherichia coli" for consideration as a Research Article by PLOS Biology.

Your manuscript has now been evaluated by the PLOS Biology editorial staff as well as by an academic editor with relevant expertise and I am writing to let you know that we would like to send your submission out for external peer review. However, our academic editor did find the manuscript in its current version confusing. A lot of the data deal with inhibition of E Coli growth when the bacteria are inside mammalian cells. However, you present data also that MAIT cells can kill free living E coli when the bacteria are not inside other cells in the absence of infected mammalian cells. On the other hand the diagram in Fig 6 at the end of the manuscript suggests that the MAIT cells first have to be stimulated by bacteria infected cells and then the MAIT cells secrete proteins which will kill free living coli. We hope you will clear up this confusion before review. 

Before we can send your manuscript to reviewers, we need you to complete your submission by providing the metadata that is required for full assessment. To this end, please login to Editorial Manager where you will find the paper in the 'Submissions Needing Revisions' folder on your homepage. Please click 'Revise Submission' from the Action Links and complete all additional questions in the submission questionnaire.

Please re-submit your manuscript within two working days, i.e. by Jan 11 2020 11:59PM.

Kind regards,

Di Jiang

PLOS Biology

---

## [Decision Letter · Decision Letter 1]

13 Feb 2020

Dear Dr Leeansyah,

Thank you very much for submitting your manuscript "Human MAIT cell cytolytic effector proteins synergize to overcome carbapenem resistance in Escherichia coli" for consideration as a Research Article at PLOS Biology. Your manuscript has been evaluated by the PLOS Biology editors, an Academic Editor with relevant expertise, and by three independent reviewers.

In light of the reviews (below), we will not be able to accept the current version of the manuscript, but we would welcome re-submission of a much-revised version that addresses all of the reviewers' comments. In addition to the concerns raised by the reviewers, our Academic Editor would like you to clarify whether the E coli you are studying are inside or outside the cells in each assay. If you think the bacteria are sometimes intracellular and sometimes extracellular, please provide a discussion on whether the action of MAIT cells is different in the two cases. We cannot make any decision about publication until we have seen the revised manuscript and your response to the reviewers' comments. Your revised manuscript is also likely to be sent for further evaluation by the reviewers.

We expect to receive your revised manuscript within 2 months. 

**IMPORTANT - SUBMITTING YOUR REVISION**

*Re-submission Checklist*

*Published Peer Review*

*PLOS Data Policy*

*Blot and Gel Data Policy*

Sincerely,

Di Jiang

PLOS Biology

REVIEWS:

Reviewer #1: 

The authors have shown that MAIT cells mediate MR1-restricted antimicrobial activity against clinical E. coli isolates in a manner dependent on the activity of cytolytic proteins, but independent of production of pro-inflammatory cytokines or induction of apoptosis in infected cells. The work has been well-planned and executed and is aimed at expanding the salubrious roles of MAIT cells across to clinical microbiology. 

What was the volume of blood drawn from donors? The purity of MAIT cells is a key concern as 30% of ancilliary cells could still play a role in overcoming MAIT cell functions or atleast impacting the outcome of the findings, for instance sourced from IFNγ, TNF, and IL-17A, pro-inflammatory cytokines. The authors must ensure to improve the purity of MAIT cells used in the experiments. Further, did the authors conduct any baseline investigations on the donors for recruiting into the study? There must be some inclusion and exclusion criteria before recruiting into the study.

The authors have not shown any data on MAIT cell activation markers or proof for MAIT proliferation (i.e. Ki67).

Basic microbiological data on the isolates of E.coli tested may be provided as supplementary data, and the authors should have used a carbapenem sensitive E.coli to prove the beneficial effect of MAIT cells on carbapenem-resistant E.coli. How do the authors justify carbapenem resistance in the isolate tested? 

Reviewer #2: In this manuscript, Boulouis and colleagues describe a possible mechanism underscoring the antimicrobial activity of MAIT cells, based on the cytolytic proteins granulysin and granzyme B, released by activated MAIT cells in an MR1 dependent manner.

The paper is well written, the experiments are well controlled and the data are convincing. This is an important study that extends our understanding of the biological functions of MAIT cells, and suggests way of potentially harnessing these cells for clinical benefit, during antibiotic resistant infections.

The main limitation of this study is that the antimicrobial effect of MAIT cell supernatants is only observed upon culturing the cells for two weeks, and although this is well discussed and acknowledged, it has not been investigated whether tissue resident MAIT cells (ie in the gastrointestinal tract) would express higher levels of granzyme B and granulysin or whether upon infection, in vivo, the kinetics of GrzB and Gly acquisition are faster.

I have a few minor comments:

1. The legend of Figure 1 G-J is not clear: the text talks about cells activated by 5OPRU or E coli but this is not clear from the graphs; furthermore, it is not clear how the heat map in panel J is derived: are these FACS data?

2. Figure S2E is misleading, if compared to S2D: although the colour scale reflects the MFI values, it gives the false impression that some markers are not expressed on D0, which is clearly not the case for Granzyme A and perforin. Why was Granzyme K not measured?

3. In Figure 1A and S2F it would be good to have the values of MAIT cell expression of the indicated proteins shown as additional overlay. Also, the authors should show the gating strategy used to discriminate between Hela and MAIT cells.

4. In Figure 3A, IL17 secretion by MAIT cells is very low and without negative controls (unpulsed target cells for example), should be interpreted with caution.

5. Figure 3: It is not clear why in panels C-E MR1 overexpressing 293T cells were used. The transition from 293T (panels C-E) to Hela is also confusing. Lastly, what is the difference between panels D and E?

6. Figure 4: in panel D, at 2 hours, the authors detect a significant difference in bacterial counts, upon exposure to activated MAIT supernatants. This, however, is not reflected in the time curves in panels G, H and in Figure S5A and C.

7. The text mentions (page 14) expression of GrzA and Prf in figure 5I, but this is only shown in S7A and B.

8. Figure S1A, is the y axis correct? Should it not be % of CD107+ cells, without MAIT?

9. Figure S3K, the colour legend is missing.

10. Colours in the growth curves (S5A, S6K) are often difficult to distinguish, perhaps a thicker line could be helpful.

11. Why the supernatant of 293T and 5OPRU (but no MAIT cells) has an anti-bacterial effect in Fig S5F?

12. The dual labelling of bacteria with Sytox green and Syto62 is very elegant, but from the text I had the impression bacteria were labelled with Syto62 before exposure to MAIT supernatants and carbapenem, hence it was not clear why MFI of Syto62 was also increasing. This was clear after reading the methods, so it would help if the main text or the legend also explained the experiment.

Reviewer #3: 

GENERAL COMMENT: 

This paper entitled "Human MAIT cell cytolytic effector proteins synergize to overcome carbapenem resistance in Escherichia coli" by Boulouis and colleagues presents results to address the mechanisms behind the antimicrobial activity of circulating human MAIT cells and explores the ability of MAIT cells to target multidrug resistant E coli and the mechanism that may be mediating this activity. 

Given that MAIT cells are notably found to be much more sensitive to several bacterial infections there is considerable interest in understanding the mechanisms that drive MAIT cell activity. This study demonstrates key effector pathways (granulysin and Grnazyme B, that are involved in MAIT cell activity against both cell associated and free living drug sensitive and resistant bacteria (in this case E Coli strains).

Strengths:

The study is a well-designed study testing several mechanisms behind human peripheral MAIT cells antimicrobial activity. The authors have evaluated key molecular pathways and include numerous controls, in comparison to non MAIT cells and pharmacological inhibitors to determine the precise role of various lytic mediators in MAIT cell activity. The report presents very relevant findings particularly highlighting the unique importance of granluysin (which has been unclear - see PLoS Pathog. 2013 Oct; 9(10): e1003681.) using a variety of assays. The approach to address the mechanism of MAIT cells activity against carbapenem AMR bacteria is intriguing and novel. Overall the manuscript has well presented data however suggestions for improvement should be considered. 

SPECIFIC COMMENTS:

A very detailed assessment of MAIT cells antibacterial activity is presented by the authors and the results are interesting. The growing threat to public health includes the rapid spread of CRE into the community and the provocative results on the role MAIT cells play in disarming carbapenem activity is a future option to be considered based on the data presented. Limitations are somewhat acknowledged as in vivo studies are lacking to validate some of the findings and this can be enhanced in the manuscript. Nevertheless, the authors reveal interesting findings on MAIT cell antimicrobial activity with clinical relevance and well-presented model in Fig 6. 

1. Previous studies have shown that MAIT cell require a cytolityic effectors and utilize the granzyme mediated pathways for killing (Kurioka et al Mucosal Immunol. 2015;8(2):429-40. So it is unclear on the novel findings presented in this report. 

2. The mechanism underlying antibacterial MAIT cell killing is comprehensive however the role of FAS/FASL or other contact dependent pathways was not determined. Co-culture transwell studies would clarify this further.

3. Fig 1G - would polyclonal stimulation rather that cytokine driven MAIT cell activation control bacterial loads? 

4. Unclear in Fig 2 if non MAIT cells were tested as controls

5. Figre 5 D and F are not convincing and could be a floor effect.

MINOR COMMENTS:

1. Fig 1 H. HELA/MAIT cells. Some of the descriptions are missing for the - - + third line data.

2. Fig 3A the color shading box description should be included as in Fig 3B

---

## [Editor Report · Decision Letter 2]

24 Apr 2020

Dear Dr Leeansyah,

Thank you for submitting your revised Research Article entitled "Human MAIT cell cytolytic effector proteins synergize to overcome carbapenem resistance in Escherichia coli" for publication in PLOS Biology. Our academic editor has assessed your revisions. We're delighted to let you know that we're now editorially satisfied with your manuscript. 

Before we can formally accept your paper and consider it "in press", we also need to ensure that your article conforms to our guidelines. A member of our team will be in touch shortly with a set of requests. As we can't proceed until these requirements are met, your swift response will help prevent delays to publication. Please also make sure to address the data and other policy-related requests noted at the end of this email.

*Copyediting*

*Published Peer Review History*

*Early Version*

*Submitting Your Revision*

Sincerely,

Di Jiang, PhD

Associate Editor

PLOS Biology

ETHICS STATEMENT:

-- Please create a separate subsection of Ethics Statement and places it in the beginning of the Methods section. Please make sure it includes all relevant information described below.

-- Please include the full name of the IACUC/ethics committee that reviewed and approved the animal care and use protocol/permit/project license. Please also include an approval number.

-- Please include the specific national or international regulations/guidelines to which your animal care and use protocol adhered. Please note that institutional or accreditation organization guidelines (such as AAALAC) do not meet this requirement.

-- Please include information about the form of consent (written/oral) given for research involving human participants. All research involving human participants must have been approved by the authors' Institutional Review Board (IRB) or an equivalent committee, and all clinical investigation must have been conducted according to the principles expressed in the Declaration of Helsinki.

DATA POLICY:

-- Regardless of the method selected, please ensure that you provide the individual numerical values that underlie the summary data displayed in the following figure panels as they are essential for readers to assess your analysis and to reproduce it: Figs 1B-L, 2B-HJ-L, 3ABD-F, 4ACDF-H, 5A-FHK, S1EH-K, S2BCEGI,S3A-IKNO, S4CF-HK, S5A-DFH, S6A-GI-L. NOTE: the numerical data provided should include all replicates AND the way in which the plotted mean and errors were derived (it should not present only the mean/average values).

-- Please provide an editor/reviewer key/token for Bioproject PRJNA577535 so we can check it before acceptance.

---

## [Editor Report · Decision Letter 3]

18 May 2020

Dear Dr Leeansyah,

On behalf of my colleagues and the Academic Editor, Philippa Marrack, I am pleased to inform you that we will be delighted to publish your Research Article in PLOS Biology. 

Early Version

PRESS 

Kind regards,

Alice Musson

Publishing Editor, 

PLOS Biology

on behalf of

Di Jiang,

Associate Editor

PLOS Biology